rsos.royalsocietypublishing.org

energy/ocean engineering/fluid mechanics

tidal power, renewable energy, resource assessment, uncertainty, bottom friction, bed roughness

**Author for correspondence:**
M. J. Kreitmair
e-mail: m.kreitmair@ed.ac.uk

# The effect of uncertain bottom friction on estimates of tidal current power

M. J. Kreitmair[1], S. Draper[2], A. G. L. Borthwick[1] and T. S. van den Bremer[1,3]

[1]School of Engineering, University of Edinburgh, Edinburgh EH9 3FB, UK
[2]Faculty of Engineering, Computing and Mathematics, The University of Western Australia, Crawley Western Australia 6009, Australia
[3]Department of Engineering Science, University of Oxford, Oxford OX1 3PJ, UK

 MJK, 0000-0001-8476-0063

Uncertainty affects estimates of the power potential of tidal currents, resulting in large ranges in values reported for a given site, such as the Pentland Firth, UK. We examine the role of bottom friction, one of the most important sources of uncertainty. We do so by using perturbation methods to find the leading-order effect of bottom friction uncertainty in theoretical models by Garrett & Cummins (2005 *Proc. R. Soc. A* **461**, 2563–2572. (doi:10.1098/rspa.2005.1494); 2013 *J. Fluid Mech.* **714**, 634–643. (doi:10.1017/jfm.2012.515)) and Vennell (2010 *J. Fluid Mech.* **671**, 587–604. (doi:10.1017/S0022112010006191)), which consider quasi-steady flow in a channel completely spanned by tidal turbines, a similar channel but retaining the inertial term, and a circular turbine farm in laterally unconfined flow. We find that bottom friction uncertainty acts to increase estimates of expected power in a fully spanned channel, but generally has the reverse effect in laterally unconfined farms. The optimal number of turbines, accounting for bottom friction uncertainty, is lower for a fully spanned channel and higher in laterally unconfined farms. We estimate the typical magnitude of bottom friction uncertainty, which suggests that the effect on estimates of expected power lies in the range −5 to +30%, but is probably small for deep channels such as the Pentland Firth (5–10%). In such a channel, the uncertainty in power estimates due to bottom friction uncertainty remains considerable, and we estimate a relative standard deviation of 30%, increasing to 50% for small channels.

## 1. Introduction

Over the past decade, rapid advances have occurred in methods used to model tidal stream resource and to optimize its extraction allowing for the feedback effect between energy removal and

natural flow conditions [1]. At coastal scale, most tide hydrodynamic models are based on the nonlinear shallow water equations. In these models, uncertainties arise from several sources, including the inexact specification of the natural environment (due to lack of accurate field data on the tidal velocity field, turbulence, large-scale eddying motions, etc.), the physical model parameters (bed roughness, bathymetry, boundary and initial conditions, etc.), model assumptions (rigid-lid approximation, requirement of low Froude number, etc.) and numerical parameters (grid resolution, time step, depth-averaged instead of three-dimensional models, etc.). Combined, these uncertainties can give rise to considerable discrepancy between different power estimates for a given site. For example, predictions of the average power available from the Pentland Firth, UK, one of the most promising sites for tidal stream energy extraction in the world, span more than an order of magnitude (from 0.62 GW [2] to 9 GW [3]), with little consensus as to the true power potential [4].

Of the sources of uncertainty listed above, the bed friction coefficient $C_0$ is particularly important. In practice, this parameter is often used to tune numerical models based on the shallow water equations, so that they predict water levels and velocity vectors in agreement with observations at relatively sparse spatial locations. Various researchers have carried out sensitivity analyses for different values of drag coefficient applied uniformly throughout the domain. For example, in a power resource assessment of the Pentland Firth, Adcock *et al.* [5] examined the sensitivity of tidal stream power estimates to the value of bed friction coefficient $C_0$ in the range $[0.0025, 0.001]$, applied uniformly through the flow domain. Adcock *et al.* found that the available power reduced as $C_0$ increased. However, the average power determined from the numerical model over the range of values of $C_0$ considered was greater than the power calculated using the average value of $C_0$. That is, the dependence of power on $C_0$ is nonlinear. In addition, Adcock *et al.* found that no single value of $C_0$ applied throughout the modelled domain produced results which matched the field measurements of both tidal phase and current magnitude for the Pentland Firth, and settled on a value of $C_0 = 0.005$ in a compromise. In a similar study, Gillibrand *et al.* [6] varied $C_0$ from 0.002 to 0.010 and found a constant value of $C_0 = 0.004$ (again applied uniformly throughout the domain) gave best agreement, while acknowledging the significant spatial heterogeneity of the seabed.

An alternative view is that the bed friction coefficient should be hydraulically correct in terms of the boundary layer dynamics and not treated simply as a tuning parameter [7]. Soulsby [8] lists a range of values ($C_0 \in [0.0011, 0.0043]$ for silt-sand to rippled sand) that could be applied to different marine bed surfaces, and which deal with skin friction, form drag and turbulence. In short, there is a lack of agreement as to which bed friction values should be applied. Culley *et al.* [9] performed a sensitivity analysis of estimated power from an optimized tidal farm in the Inner Sound of the Pentland Firth, which highlighted the significant influence of the value of the bottom friction coefficient on the numerical results. The estimated power reduced as the bed roughness increased near the farm, and, at sufficiently high values of local bed friction, the flow began to bypass the farm along paths of lower frictional resistance.

This paper aims to address how uncertainty in the parametrization of bed friction affects estimates of extractable power in different analytic models for tidal energy extraction in which turbines are represented as either local or global enhanced bed roughness. Insight into the effect of the underlying physical assumptions on uncertainty propagation is developed by considering closed-form solutions for power dissipated as predicted by three analytic models of tidal power assessment. The first model is that of Garrett & Cummins [10] (henceforth GC05), who derive an analytic solution for quasi-steady flow in a channel spanned completely by tidal turbines. Second, we explore the impact of retaining inertia by examining the solution to the same governing equation by Vennell [11] (henceforth V10). V10 is able to include inertia in a closed-form solution by making further approximations (see appendix of V10). Third and finally, we examine the effect of flow diversion around the turbines by considering Garrett & Cummins [12] (henceforth GC13), who consider a circular turbine farm in laterally unconfined flow. Analytic solutions of these types have been shown to give predictions in satisfactory agreement with results from numerical models [13,14]. We introduce uncertainty in the value of background roughness coefficient in these models and use perturbation methods to identify the leading-order effect of this uncertainty on the expected power dissipated by the turbines and the optimal channel design. Using our best estimate for the magnitude of the uncertainty in background roughness coefficient, we provide quantitative estimates of the effects of uncertainty on expected power dissipated and optimal channel design.

This paper is laid out as follows. In §2, after a brief review of each model, we introduce uncertainty into the three theoretical models (GC05, V10 and GC13) and obtain leading-order estimates of the effect of uncertainty using perturbation methods. In §3, we obtain a best estimate for the relative standard

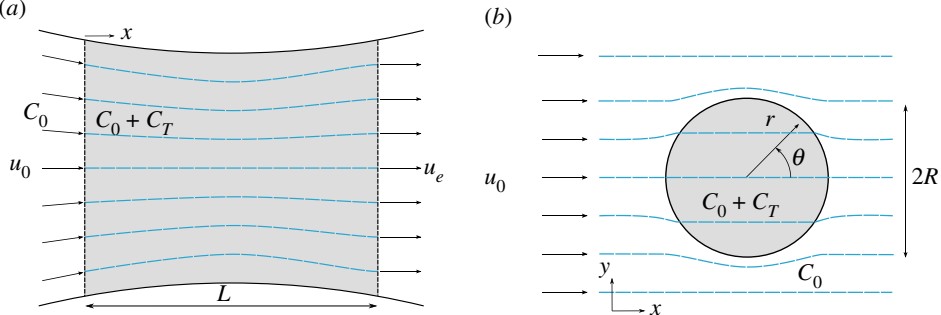

**Figure 1.** (a) Definition sketches for the fully spanned channel models of GC05 and V10 (adapted from [10]) and (b) the laterally unconfined model of GC13, where the shaded area is the region of increased bed friction ($C_0 + C_T$, in which $C_0$ is the background friction) representing the turbine farm of radius $R$ with a uniform upstream velocity of $u_0$ in the x-direction. Streamlines are shown as blue dashed lines.

rsos.royalsocietypublishing.org   R. Soc. open sci. **6**: 180941

deviation of the bed roughness coefficient (the ratio of the standard deviation in the value of $C_0$ to its mean). Using this calibration and our leading-order solutions, we examine the effects of uncertainty in §4 and compare the three models. Finally, we draw conclusions in §5.

# 2. Introducing uncertainty in theoretical models

## 2.1. Fully spanned channel (GC05 and V10)

In the model of GC05 (figure 1a) power is extracted from a channel of length $L$ and depth $h$ connecting two large bodies of water, by means of a fence of turbines that fully spans the cross-section of the channel. The flow is driven in the simplest case by a sinusoidal tide producing a head difference between the ends of the channel, of amplitude $a$ and angular frequency $\omega$. Water is drawn in smoothly at speed $u_0$ at the entrance of the channel and exits as a jet at speed $u_e$. Furthermore, the channel is assumed sufficiently short compared with the tidal wavelength that the volume flux $Q$ is independent of distance along the channel $x$. These assumptions allow integration of the one-dimensional shallow water momentum equation along the length of the channel to give (GC05)

$$\gamma \frac{\mathrm{d}Q}{\mathrm{d}t} - ga \cos{(\omega t)} = -(\delta_0 + \delta_T)|Q|Q. \tag{2.1}$$

Here $\gamma = \int_0^L A^{-1}\mathrm{d}x$ is a geometric factor taking into account the varying cross-sectional area $A$ of the channel, $t$ is time, and the term $ga \cos(\omega t)$ is the driving pressure force due to the tide where $g$ is gravity. On the right-hand side, $\delta_0 = \int_0^L C_0(hA^2)^{-1}\mathrm{d}x + (1/2)A_e^{-2}$ accounts for the friction due to a given bed roughness coefficient $C_0$ and the velocity head loss at the channel exit where the cross-sectional area is $A_e$, and $\delta_T = \int_0^L C_T A^{-2}\mathrm{d}x$ represents the energy dissipated due to power extraction, with turbines represented by a distributed roughness coefficient $C_T$. By introducing the non-dimensional variables $t' = \omega t$, $Q' = Q\omega\gamma/(ga)$, $\lambda_0 = ga\delta_0/(\gamma\omega)^2$ and $\lambda_T = ga\delta_T/(\gamma\omega)^2$, GC05 obtain the expression

$$\frac{\mathrm{d}Q'}{\mathrm{d}t'} - \cos{(t')} = -(\lambda_0 + \lambda_T)|Q'|Q'. \tag{2.2}$$

The value of the parameter $\lambda_0$ determines the dynamic balance within the channel. It represents the ratio of the combination of the natural drag losses and exit separation to acceleration in the channel, normalized by the driving amplitude [10]. Large values of $\lambda_0$ describe channels dominated by background friction and exit separation, i.e. shallow, short channels in which the flow may be considered to be quasi-steady. Small values of $\lambda_0$ correspond to channels in the inertial limit as would be the case for deep, long channels. The power dissipated by the turbines is given by multiplication of the turbine drag term by the mass flow rate, i.e. $P = \rho\delta_T|Q|Q^2$, where $\rho$ is the fluid density. The average power extracted by the turbines over a tidal cycle is then $\bar{P} = \rho\delta_T\overline{|Q|Q^2} = \rho(ga)^2(\gamma\omega)^{-1}\lambda_T\overline{|Q'|Q'^2}$, where the overline notation indicates time-averaging over the tidal period. The non-dimensional flow rate $Q'$ is found by solving (2.2) and is, for a given head difference, a function of time and the total drag in the channel, i.e. $Q'(t', \lambda_0 + \lambda_T)$.

### 2.1.1. The quasi-steady limit (GC05)

GC05 derive an analytical solution for the average power in the quasi-steady state limit, i.e. for large values of $\lambda_0$. In this limit, the acceleration term in (2.2) may be neglected and the non-dimensional volumetric flux may then be approximated by $|Q'| = (\lambda_0 + \lambda_T)^{-1/2}|\cos t'|^{1/2}$. The corresponding average power produced by the turbines becomes

$$P_{\text{GC05}} = P_0 \frac{\lambda_T}{(\lambda_0 + \lambda_T)^{3/2}}, \tag{2.3}$$

where $P_0 = \beta_2 \rho (ga)^2/(\gamma\omega)$ is the dimensional multiplier for the power and $\beta_2 = \overline{|\cos t'|^{3/2}} \approx 0.56$ accounts for time-varying head difference (and the subscript 2 denotes quadratic friction).

To introduce uncertainty in background friction, we express $\lambda_0$ as a random variable with an expected value of $\mu_{\lambda_0}$ and random, zero-mean fluctuation $\Delta\lambda_0$ about this value, such that $\lambda_0 = \mu_{\lambda 0} + \Delta\lambda_0$. Provided the fluctuation is small compared with the mean, the power produced by the turbines may be expressed in terms of $\lambda_0$ by expanding (2.3) as a Taylor series in $\Delta\lambda_0$ about the deterministic case ($\Delta\lambda_0 = 0$) as

$$\frac{1}{P_0}P_{\text{GC05}} = \frac{\lambda_T}{(\mu_{\lambda_0} + \lambda_T)^{3/2}} - \frac{3}{2}\frac{\lambda_T}{(\mu_{\lambda_0} + \lambda_T)^{5/2}}\Delta\lambda_0 + \frac{15}{8}\frac{\lambda_T}{(\mu_{\lambda_0} + \lambda_T)^{7/2}}\Delta\lambda_0^2 + \mathcal{O}(\Delta\lambda_0^3). \tag{2.4}$$

Higher-order terms are neglected in the series, which converges for sufficiently small $\Delta\lambda_0$. Only leading-order effects resulting from the mean and standard deviation in the bed roughness probability density function are considered.

### 2.1.1.1. Expected power

Applying the expectation operator, the expected power extracted, correct to second order in $\Delta\lambda_0$, is given by

$$\frac{1}{P_0}E[P_{\text{GC05}}] = \frac{\lambda_T}{(\mu_{\lambda_0} + \lambda_T)^{3/2}} + \frac{15}{8}\frac{\lambda_T}{(\mu_{\lambda_0} + \lambda_T)^{7/2}}\sigma_{\lambda_0}^2 + \mathcal{O}(E[\Delta\lambda_0^3]), \tag{2.5}$$

where $\sigma_{\lambda_0}^2 = E[\Delta\lambda_0^2]$ is the variance in background friction parameter. The second term of the series (2.4) vanishes as the random fluctuation $\Delta\lambda_0$ is symmetric about the mean. The first term in the expansion is simply the deterministic power removed by the turbines in a channel (2.3) at the mean background friction parameter $\mu_{\lambda_0}$. The second term is a stochastic correction to the power resulting from considering a distribution of $\lambda_0$ values that are spread about the mean $\mu_{\lambda_0}$ with a standard deviation of $\sigma_{\lambda_0}$. We do not consider higher-order terms that take account of corrections due to further moments of the probability density function, such as skewness and kurtosis.

The expected power (2.5) is shown as a function of turbine drag parameter $\lambda_T$ in figure 2 for two channels with different values of the mean background friction parameter $\mu_{\lambda_0}$. The first channel, with $\mu_{\lambda_0} = 1.0$, corresponds to a large and deep channel, and the second, with $\mu_{\lambda_0} = 4.5$, to a small channel with a high flow velocity [15]. It is clear that, regardless of the mean channel drag parameter or the value of turbine drag, uncertainty in $\lambda_0$ acts to increase expected power (dashed and dot-dashed lines) from that calculated using the deterministic model (continuous lines) and more so for greater $\sigma_{\lambda_0}$ values. This effect is greatest for $\lambda_T = 2\mu_{\lambda_0}/5$, which maximizes the second term in (2.5), but remains positive for all values of $\lambda_T$, reducing in strength as $\lambda_T$ increases (and the effect of background roughness becomes less important).

This increase in expected power is a result of the inverse relationship between power (2.3) and bed friction parameter $\lambda_0$. Neglecting the inertial term in (2.2) (by assumption of the quasi-steady limit) requires that the head difference driving the flow is balanced solely by dissipation due to the total channel drag $\lambda_{\text{tot}} = \lambda_0 + \lambda_T$. Hence the flow rate $Q$ is inversely proportional to $\lambda_{\text{tot}}$ and, for a given driving head, $Q$ must grow increasingly fast as total channel drag reduces, i.e. $\partial^2 Q/\partial\lambda_{\text{tot}}^2 > 0$. Consequently, a small reduction in bed roughness parameter away from the mean $\Delta\lambda_0^- < 0$ results in dissipation of a greater amount of power by the turbines, $\Delta P_{\text{GC05}}^- > 0$. Similarly, a small increase in bed roughness parameter of the same magnitude $\Delta\lambda_0^+ > 0$ yields reduction by an amount of power $\Delta P_{\text{GC05}}^+ < 0$ smaller than before, i.e. $|\Delta P_{\text{GC05}}^+| < \Delta P_{\text{GC05}}^-$. Assuming a symmetric probability density function for $\lambda_0$, the expected power, given by $E[P_{\text{GC05}}] = P_{\text{GC05}}(\lambda_0 = \mu_{\lambda_0}) + (\Delta P_{\text{GC05}}^- + \Delta P_{\text{GC05}}^+)/2$, is then necessarily greater than the deterministic power. In other words, due to the dynamic balance between driving head and channel drag, the power curve has a positive second derivative with respect to the channel bed roughness $\lambda_0$. This convexity results in an asymmetric power dissipation for symmetric perturbations in $\lambda_0$ and thus an increase in the expected power (cf. Jensen's inequality, which states that a convex transformation of the mean of a random variable is less than the mean of the convex transformation of the variable).

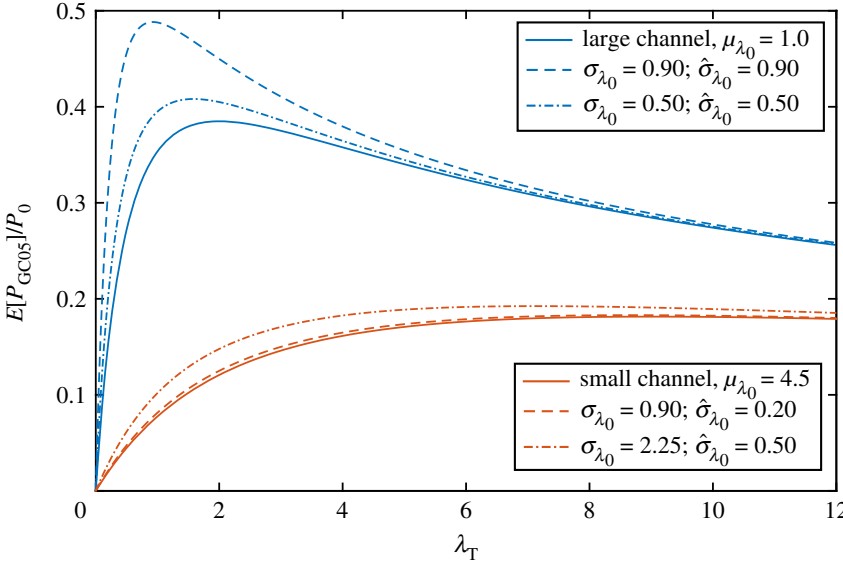

**Figure 2.** Expected power produced by turbines in two fully spanned tidal channels (GC05) with mean background friction parameter values of $\mu_{\lambda_0} = 1.0$ (representative of a large and deep channel) and $\mu_{\lambda_0} = 4.5$ (representative of a small channel). Power from equivalent deterministic channels is shown as solid lines. The dashed lines show the expected power from the two channels at the same value of standard deviation in background friction parameter $\sigma_{\lambda_0} = 0.90$. The dot-dashed lines have the same value of relative standard deviation $\hat{\sigma}_{\lambda_0} = \sigma_{\lambda_0}/\mu_{\lambda_0} = 0.50$.

### 2.1.1.2. Optimal turbine drag

In addition to a change in expected power, figure 2 also shows a shift in optimal turbine drag due to uncertainty. In the absence of uncertainty, this optimum occurs at a value of turbine drag that is twice the mean background friction parameter: $\lambda_{T\text{det}}^* = 2\mu_{\lambda 0}$. However, with increasing $\sigma_{\lambda_0}$ the maximum shifts to lower values of $\lambda_T$. An analytical expression for the optimal turbine drag $\lambda_{T\text{stoch}}^*$ may be found by maximizing (2.5) with respect to $\lambda_T$ such that

$$\lambda_{T\text{stoch}}^* = 2\mu_{\lambda 0}\left[1 - \frac{5}{6}\hat{\sigma}_\lambda^2\right] + \mathcal{O}(E[\Delta\lambda_0^3]) \quad \text{with } \hat{\sigma}_{\lambda_0} = \frac{\sigma_{\lambda_0}}{\mu_{\lambda_0}}. \tag{2.6}$$

The optimal turbine drag reduces linearly with the variance of the bed friction parameter. This may be understood by perturbing around the deterministic optimum, so that $\lambda_T^* = \lambda_{T\text{det}}^* + \Delta\lambda_T^*$. Along the optimum, we have $\partial E[P_{\text{GC05}}]/\partial\lambda_T = 0$. Expanding this identity around $\lambda_{T\text{det}}^*$ gives, after some manipulation

$$\Delta\lambda_T^* = -\frac{1}{2}\frac{P_{\lambda_0\lambda_0\lambda_T}(\mu_{\lambda 0}, \lambda_{T\text{det}}^*)}{P_{\lambda_T\lambda_T}(\mu_{\lambda 0}, \lambda_{T\text{det}}^*)}\sigma_\lambda^2, \tag{2.7}$$

where $P$ corresponds to $P_{\text{GC05}}$ and the subscripts denote differentiation. The change in optimal turbine drag $\Delta\lambda_T^*$ depends on the sign of $P_{\lambda_0\lambda_0\lambda_T}$ (the change in the convexity of the power curve with turbine drag) and the sign of $P_{\lambda_T\lambda_T}$ (the convexity of the power with respect to turbine drag) calculated at the deterministic optimum to leading order of approximation. We have $P_{\lambda_T\lambda_T} < 0$ because of the maximum. The effect of reducing $\lambda_T$ is to lower the total channel drag, making the flow rate and hence the power more sensitive to the bed friction parameter. At lower values of $\lambda_{\text{tot}}$, the increase in power becomes relatively larger than the decrease in power for a fluctuation $\Delta\lambda_0$, and the change in the expected power increases ($\partial^3 P/\partial\lambda_{\text{tot}}^3 < 0$). It is therefore optimal in the presence of background friction uncertainty to choose a lower value of $\lambda_T$ in order to harness better the uncertain power.

### 2.1.1.3. Uncertainty in power

The variance in power, $\sigma_P^2 = E[(P - E[P])^2]$, may be evaluated using (2.3) to give to leading order

$$\sigma_P^2 = \frac{9}{4}P_0^2\frac{\lambda_T^2}{(\mu_{\lambda 0} + \lambda_T)^5}\sigma_{\lambda 0}^2 + \mathcal{O}(E[\Delta\lambda_0^3]) \tag{2.8}$$

$$= (P_{\text{GC05}}(\lambda_0 = \mu_{\lambda_0}))^2\frac{9}{4}\frac{\hat{\sigma}_{\lambda_0}^2}{(1 + \lambda_T/\mu_{\lambda_0})^2} + \mathcal{O}(E[\Delta\lambda_0^3]), \tag{2.9}$$

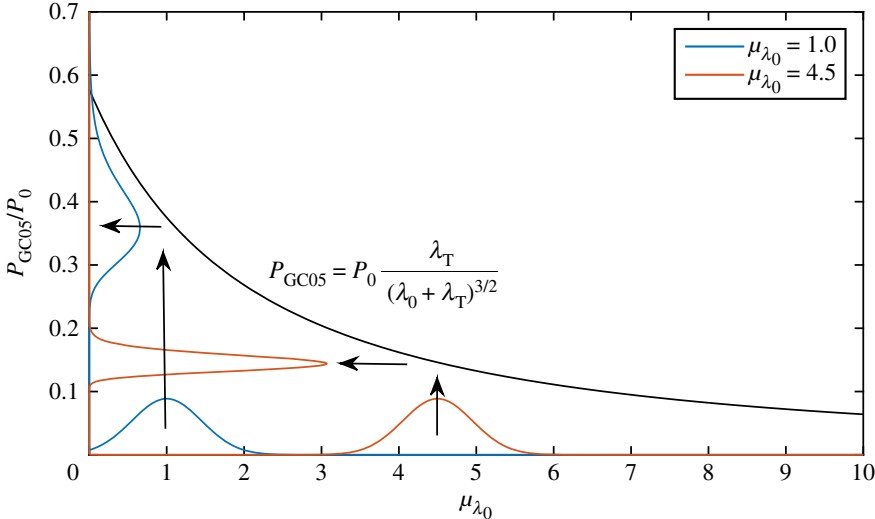

**Figure 3.** Mapping of uncertainty from background friction $\lambda_0$ to power $P_{GC05}$ via the transfer function of the power curve for a fully spanned tidal channel (GC05). For both values of the mean background friction $\mu_{\lambda_0} = 1.0$ (representative of a large and deep channel) and $\mu_{\lambda_0} = 4.5$ (representative of a small channel) the standard deviation in $\lambda_0$ is $\sigma_{\lambda_0} = 0.45$ and the turbine drag is $\lambda_T = 3.0$.

where $\hat{\sigma}_{\lambda_0} = \sigma_{\lambda_0}/\mu_{\lambda_0}$ is the relative standard deviation. The greater the total mean drag in the channel $\mu_{\lambda_0} + \lambda_T$, the smaller the standard deviation in power. This may be understood by considering the mapping of the probability density function of the background friction parameter $f_{\lambda_0}$ to the probability density function of power $f_P$: $f_P(P) = f_{\lambda_0}(\lambda_0(P))/|dP/d\lambda_0|$. Figure 3 illustrates this mapping. Probability density functions are shown for two different values of mean background friction, $\mu_{\lambda_0} = 1.0$ and $\mu_{\lambda_0} = 5.0$, for the same standard deviation of $\sigma_{\lambda_0} = 0.45$ and at a turbine drag of $\lambda_T = 3.0$. The greater the value of $\mu_{\lambda_0}$, the smaller the standard deviation in power, due to the smaller gradient in the transfer function.

We note from figure 3 that, despite symmetric input probability density functions for $\lambda_0$, the corresponding probability density functions for the power values are asymmetric. Propagation through the nonlinear transfer function has generated (positive) skewness in power. It is worth noting that the probability distribution of total channel friction is technically not allowed to have zero negative values. Because we only consider leading-order terms in uncertainty we automatically avoid the singular or complex values of power implied by zero or negative values of total channel friction.

### 2.1.2. The effect of inertia (V10)

The quasi-steady limit in the previous section applies to channels in which friction dominates inertia in the dynamic balance of the channel, i.e. in the limit of large $\lambda_0$ values, and the inertial term in (2.2) may be neglected. Relaxation of the quasi-steady assumption leads to a different behaviour of the power potential of the channel under bed roughness uncertainty. We explore the effect of retaining inertia in the channel dynamics by considering the solution presented in the appendix of Vennell [11] (V10). Therein an analytic solution is derived to an approximation of (2.2) which retains the inertial term. Furthermore, the quadratic drag term is replaced with a linear drag term which ensures the same average power is dissipated by the turbines over a tidal cycle, a process known as Lorentz linearization [16,17].

Following this approach, and assuming a sinusoidal driving tide of single frequency $\omega$ as before, the drag term $(\lambda_0 + \lambda_T)|Q'|Q'$ in (2.2), where $Q'$ is the non-dimensional flow rate, may be replaced with $KQ'$ such that $(\lambda_0 + \lambda_T)\overline{|Q'|Q'^2} = K\overline{Q'^2}$, where $Q' = Q_0'\sin(t' - \phi_Q)$ and $\phi_Q$ is the phase lag of the flow rate to the driving head difference between the ends of the channel. The coefficient $K$ may be evaluated as $K = 8(\lambda_0 + \lambda_T)Q_0'/(3\pi)$. The resulting linearized governing equation gives (V10)

$$Q_0'\cos(t' - \phi_Q) - \cos(t') \approx \frac{8}{3\pi}(\lambda_0 + \lambda_T)Q_0'^2\sin(t' - \phi_Q), \quad (2.10)$$

and may be solved to give the solutions (V10)

$$Q_0' = \frac{\left(\sqrt{4\lambda_{eq}^2 + 1} - 1\right)^{1/2}}{\sqrt{2}\lambda_{eq}} \quad \text{and} \quad \phi_Q = \tan^{-1}\left(\frac{1}{\lambda_{eq}Q_0'}\right), \quad (2.11)$$

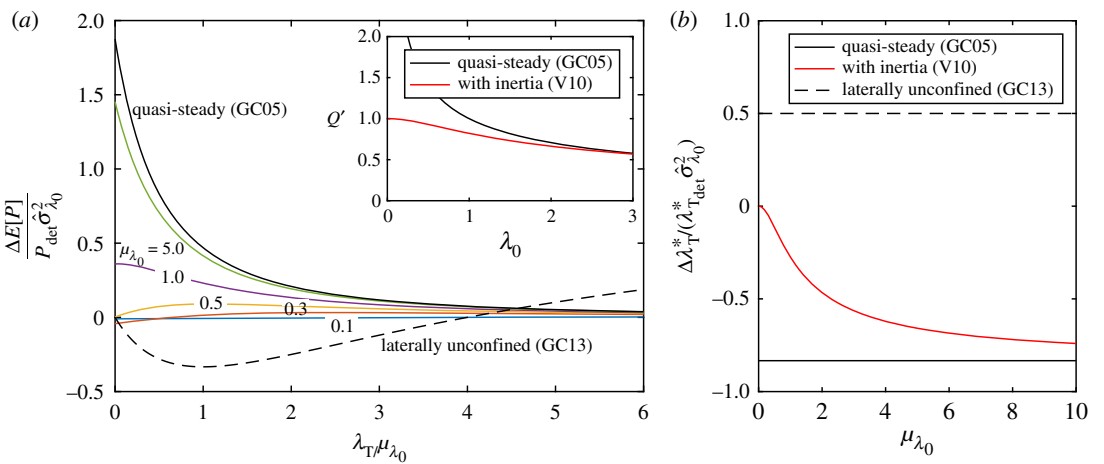

**Figure 4.** Relative change in expected power per unit relative variance in background friction $\hat{\sigma}_{\lambda_0}^2$ as a function of turbine drag scaled with mean background friction at different values of $\mu_{\lambda_0}$ (a) and relative change in optimal turbine friction in the presence of uncertainty in the background friction parameter (b) for a fully spanned tidal channel (GC05 and V10). In panel (a), for the GC05 model (black line) the lines from figure 2 reduce to the same curve upon scaling. The inset shows the non-dimensional flow rate $Q' = \omega x/(ga)Q$ as a function of channel friction parameter $\lambda_0$ for an undisturbed channel, i.e. $\lambda_T = 0$.

where the equivalent total friction parameter $\lambda_{\text{eq}} \equiv 8(\lambda_0 + \lambda_T)/(3\pi)$. Finally, the power produced by the turbines for this model is given by

$$P_{\text{V10}} = \frac{P_0}{\beta_2} \lambda_T \frac{8}{3\pi} Q_0'^3 \overline{\sin{(t' + \phi_Q)^2}} = \frac{P_0}{\beta_2} \frac{4}{3\pi} \lambda_T \frac{\left(\sqrt{4\lambda_{\text{eq}}^2 + 1} - 1\right)^{3/2}}{(\sqrt{2}\lambda_{\text{eq}})^3}. \tag{2.12}$$

As before, the magnitude of $\lambda_0$ defines the dynamic balance in the channel: small values indicate a channel that is dominated by inertia, whereas large values of $\lambda_0$ imply that background friction dominates. In the limit $\lambda_0 \to \infty$ (the quasi-steady limit), we recover GC05 (2.3) and in the limit of $\lambda_0 \to 0$ (the inertial limit) we obtain

$$P_{\text{V10}} \to \frac{P_0}{\beta_2} \frac{\sqrt{3\pi}}{256\sqrt{2}} \frac{\left(\sqrt{9\pi^2 + 256\lambda_T^2} - 3\pi\right)^{3/2}}{\lambda_T^2} \quad \text{as } \lambda_0 \to 0, \tag{2.13}$$

which is independent of $\lambda_0$.

### 2.1.2.1. Expected power

For general values of $\lambda_0$, we adopt the same approach as for the model of GC05 in the previous section. We expand (2.11) in terms of the background friction parameter and apply the expectation operator to derive an expression for the expected power $E[P_{\text{V10}}]$. The resulting equations are cumbersome, do not lend additional insight and are hence not shown here, but given in appendix A. Instead, figure 4a shows the change in the expected power, correct to second order in $\sigma_{\lambda_0}$, per unit relative variance $\sigma_{\lambda_0}^2$ (also known as the coefficient of variation) in $\lambda_0$ as a function of the turbine drag scaled by the mean background friction parameter. For increasing $\mu_{\lambda_0}$, the effect of background friction uncertainty approaches that of GC05, as illustrated by the different colour curves.

Furthermore, as the value of mean background friction $\mu_{\lambda_0}$ is reduced, the change in expected power drops to zero (see the line for $\mu_{\lambda_0} = 0.1$), reflecting the independence of power from background friction in the limit of small $\lambda_0$. In short, figure 4a indicates that inertia reduces the effect of uncertainty on expected power. The transition from the quasi-steady to the inertial limit can be non-monotonic. For channels with background friction $\mu_{\lambda_0} > \lambda_{\text{transition}}$ with $\lambda_{\text{transition}} = 0.495$, the change in expected power is positive for all values of $\lambda_T$ and the flow dynamics are dominated by the effect of the total channel drag $\lambda_{\text{tot}} = \lambda_0 + \lambda_T$. For values of $\mu_{\lambda_0}$ below $\lambda_{\text{transition}}$, the change in expected power becomes negative for values of turbine friction given by $\lambda_T < \lambda_{\text{transition}} - \mu_{\lambda_0}$, as may be seen from the curves with $\mu_{\lambda_0} = 0.1$ and 0.3.

This behaviour may be understood by considering the flow rate $Q'$ as a function of bed friction parameter for an undisturbed channel, shown in the inset in figure 4a. The singular limit as $\lambda_0 \to 0$ in

rsos.royalsocietypublishing.org R. Soc. open sci. 6: 180941

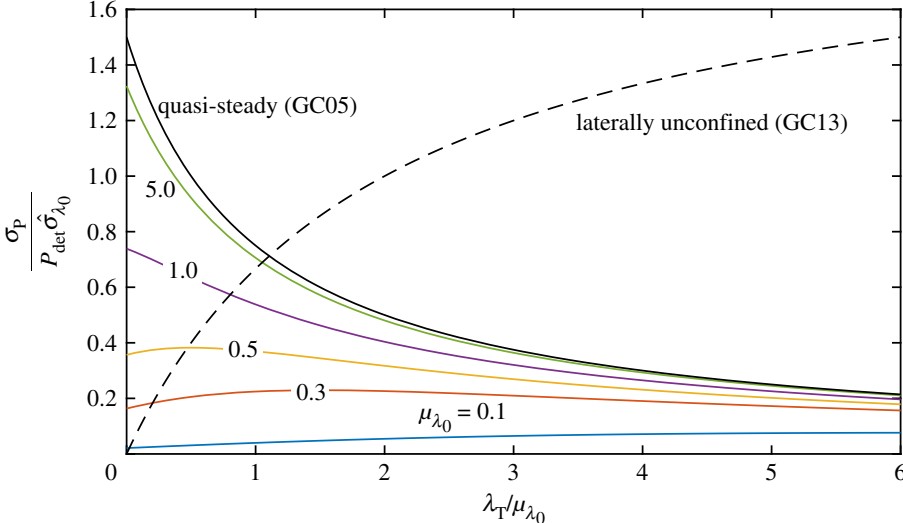

**Figure 5.** Relative standard deviation in power per unit relative standard deviation in background friction $\hat{\sigma}_{\lambda_0}$ as a function of the turbine drag scaled with mean background friction. For the model retaining inertia (V10), this is a function of mean background friction and has been plotted at different values of $\mu_{\lambda_0}$.

GC05, is avoided by inertia in V10. Owing to the bounded nature of the flow rate at low values of channel friction, a small decrease in bed roughness $\Delta\lambda_0^-$ will only slightly increase the flow rate because the channel is inertia-dominated ($\Delta P > 0$ but small). On the other hand, an increase $\Delta\lambda_0^+$ will be less affected by inertia, as the move is towards the drag-dominated regime ($\Delta P < 0$ and high). The expected power is therefore lower than in the deterministic case for $\mu_{\lambda_0} < \lambda_{\text{transition}} = 0.495$, where $\lambda_{\text{transition}}$ demarcates the transition between inertia-dominated and drag-dominated channels.

### 2.1.2.2. Optimal turbine drag

In the V10 model, as for the GC05 model, the optimal turbine tuning changes upon introduction of uncertainty in $\lambda_0$. Figure 4$b$ shows the relative change in the optimal turbine drag $\lambda_T^*$ per unit variance in $\lambda_0$, as a function of the mean bed friction coefficient. In the limit of zero background friction $\mu_{\lambda_0} \to 0$, the optimal turbine drag is unaffected because the flow behaviour is dominated by inertia. At very large values of the mean bed friction coefficient, the system becomes dominated by friction and the V10 model asymptotically approaches the quasi-steady limit of GC05.

### 2.1.2.3. Uncertainty in power

Compared to the quasi-steady limit (GC05), in which the relative standard deviation in power as a fraction of the standard deviation in background friction is a monotonically decreasing function of turbine drag scaled with mean background friction (cf. equation (2.8)), inertia reduces the effect of uncertainty, as illustrated in figure 5. As the dependence of power on background friction is reduced in the inertia-dominated regime, the resulting variance in power is smaller.

## 2.2. Laterally unconfined turbine farm (GC13)

For turbine farms that do not span the channel completely, not all of the flow in the channel passes through the turbines, instead part of it is diverted around the turbines as bypass flow. In such cases, bed friction acts not only to reduce the flow speed in the channel, but also to funnel the flow through the turbine farm by resisting the bypass flow. We explore these competing effects by considering the model of Garrett & Cummins [12] (GC13). In this model, energy extraction by a tidal farm is represented by a localized increase in bed roughness within a circular area of radius $R$ in a steady flow of far-field current of $u_0$ in the $x$-direction and no lateral confinement by channel walls or similar (figure 1$b$). The dynamic balance of the system is described by the shallow-water equations

$$\frac{\partial \mathbf{u}}{\partial t} + \mathbf{f} \times \mathbf{u} + \mathbf{u} \cdot \nabla \mathbf{u} + g \nabla \zeta = -\frac{C_0}{h + \zeta} |\mathbf{u}| \mathbf{u}, \tag{2.14}$$

rsos.royalsocietypublishing.org R. Soc. open sci. 6: 180941

where $f$ is Coriolis frequency $f$ multiplied by the unit vertical vector, $h$ is mean water depth, $\zeta$ is deviation of free surface from mean depth, and $C_0$ is bed-roughness coefficient associated with a quadratic drag law. If the rigid-lid approximation is made, i.e. $\zeta \ll h$, a very reasonable approximation given the local spatial scale of the turbine compared to the tidal wavelength, the accompanying continuity equation reduces to $\nabla \cdot \mathbf{u} = 0$ and the Coriolis vector vanishes from the vorticity equation (as follows). By subsequently linearizing the bottom friction, $(C_0/(h + \zeta))|\mathbf{u}|\mathbf{u} \to C_{L,0}\mathbf{u}$, and taking the curl of (2.13) the resulting vorticity equation obtained in GC13 gives

$$\frac{\partial \nabla^2 \psi}{\partial t} + J(\psi, \nabla^2 \psi) = -C_L \nabla^2 \psi - \nabla C_L \cdot \nabla \psi, \tag{2.15}$$

where $\psi$ is the streamfunction defined as $\mathbf{u} = (-\partial \psi/\partial y, \partial \psi/\partial x)$ and $J$ is the Jacobian. At steady state, neglecting the nonlinear material derivative and using polar coordinates $(r, \theta)$, the solution for the streamfunction is (GC13)

$$\psi = \begin{cases} -\left(1 - \dfrac{C_L}{C_L + 2C_{L,0}}\dfrac{R^2}{r^2}\right) u_0 r \sin \theta, & \text{for } r > R \tag{2.16} \\[2ex] -\dfrac{2C_{L,0}}{C_L + 2C_{L,0}} u_0 r \sin \theta, & \text{for } r \leq R. \tag{2.17} \end{cases}$$

where $C_{L,0}$ denotes the linear background friction ($C_L = C_{L,0}$ for $r > R$) and $C_{L,T}$ is the additional friction associated with the turbine farm ($C_L = C_{L,0} + C_{L,T}$ for $r \leq R$), as illustrated in figure 1$b$ with streamlines shown as blue dashed lines. The streamfunction within the farm (2.17) is equivalent to uniform flow in the $x$-direction at constant speed, $u_T = 2u_0 C_{L,0}/(C_{L,T} + 2C_{L,0})$. Higher background friction $C_{L,0}$ invariably has the effect of directing a larger proportion of the flow through the farm such that the flow velocity increases within the farm ($r \leq R$) with $C_{L,0}$. Power dissipated by the turbines is given by the integral over the fluid of the linear friction coefficient of the turbines $C_{L,T}$ multiplied by the square of the flow speed within the farm (CG13)

$$P_{\text{GC13}} = P_0 \lambda_T \left(\frac{\lambda_0}{\lambda_T + 2\lambda_0}\right)^2, \tag{2.18}$$

where $P_0 = 4\pi C_{0,\text{ref}}\rho\pi R^2 u_0^3$ and we have introduced the non-dimensional background friction $\lambda_0 = C_{L,0}h/C_{0,\text{ref}}u_0$ and the non-dimensional turbine friction $\lambda_T = C_{L,T}h/C_{0,\text{ref}}u_0$, which are analogous, but not equivalent to their counterparts for GC05 and V10. In order to facilitate comparison with the fully spanned channel, we have scaled $\lambda_0$ and $\lambda_T$ by a non-stochastic reference drag coefficient $C_{0,\text{ref}}$, so that typical values of $\lambda_0$ and $\lambda_T$ are $O(1)$. The deterministic power extracted is maximized at a turbine drag of $\lambda_T^* = 2\lambda_0$.

As for the previous two models, we introduce uncertainty in background friction by expressing $\lambda_0$ as a normally distributed random variable with an expected value of $\mu_{\lambda_0}$ and variance $\sigma_{\lambda_0}^2$. Provided the variation is small compared with the mean, the power produced by the turbines may be expressed in terms of $\lambda_0$ by expanding (2.15) as a Taylor series in $\Delta\lambda_0 = \lambda_0 - \mu_{\lambda_0}$ about the deterministic case ($\Delta\lambda_0 = 0$).

### 2.2.1. Expected power

Performing the Taylor series expansion and evaluating the expectation operator for the leading-order effect of uncertainty, gives

$$\frac{1}{P_0}E[P_{\text{GC13}}] = \frac{\lambda_T \mu_{\lambda_0}^2}{(\lambda_T + 2\mu_{\lambda_0})^2} + \frac{\lambda_T^2(\lambda_T - 4\mu_{\lambda_0})}{(\lambda_T + 2\mu_{\lambda_0})^4}\sigma_{\lambda_0}^2 + \mathcal{O}(E[\Delta\lambda_0^3]), \tag{2.19}$$

where the first term corresponds to the deterministic power (evaluated at mean background friction) and the second term provides a correction resulting from the background friction uncertainty. Figure 6 shows expected power as a function of turbine friction $\lambda_T$ for different values of mean background friction coefficient $\mu_{\lambda_0}$ and standard deviation $\sigma_{\lambda_0}$. Unlike the quasi-steady limit of the fully spanned channel (GC05), where the change in expected power from deterministic power is positive regardless of turbine drag, the sign of the correction term now depends on the relative magnitude of turbine drag and bed friction: for $\lambda_T < 4\lambda_0$ the expected power is reduced, and *vice versa* for $\lambda_T > 4\lambda_0$.

This non-monotonicity can be explained as follows. For sufficiently small values of background friction, power is approximately quadratic in $\lambda_0$ (because $P_{\text{GC13}} \propto u_T^2$ and $u_T \propto \lambda_0$ for $\lambda_0 \ll \lambda_T$) and a

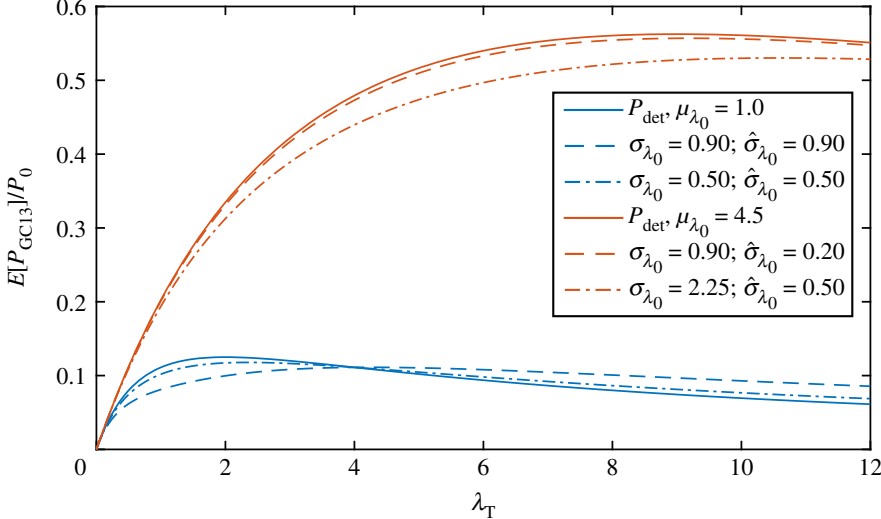

**Figure 6.** Expected power produced by a laterally unconfined turbine farm (GC13) for two scenarios with mean background friction parameter values of $\mu_{\lambda_0} = 1.0$ and $\mu_{\lambda_0} = 4.5$. Deterministic power is indicated by solid lines. The dashed lines show the expected power from the two channels at the same value of standard deviation in the background friction parameter $\sigma_{\lambda_0} = 0.90$. The dotted-dashed lines have the same value of relative standard deviation $\hat{\sigma}_{\lambda_0} = \sigma_{\lambda_0}/\mu_{\lambda_0} = 0.50$.

small increase $\lambda_0^+ > 0$ produces a greater increase in power than the reduction in power resulting from a decrease $\lambda_0^- < 0$ of the same magnitude (i.e. $P_{GC13}$ is a convex function of $\lambda_0$). Hence a net increase in expected power occurs as a result of uncertainty, as may be seen in figure 4a for large $\lambda_T/\mu_{\lambda_0}$ (corresponding to small $\mu_{\lambda_0}$). On the other hand, at large values of $\lambda_0$ a small increase in background friction has a relatively smaller effect on flow rate (cf. $u_T/u_0 \to 1$ for $\lambda_0 \gg \lambda_T$), and consequently power, than a decrease in background friction of equal magnitude. As $\lambda_0$ increases, the flow speed initially increases, but then tends towards a constant value. The decreasing rate of change of flow speed with $\lambda_0$ results in a concave dependence of power on $\lambda_0$ ($\partial^2 P_{GC13}/\partial \lambda_0^2 < 0$) for sufficiently large $\lambda_0$. This results in a net decrease in expected power. The transition between the two regimes occurs at $\lambda_T = 4\lambda_0$, as is evident from (2.16).

In a completely spanned channel (GC05), the flow rate decreases with increasing background friction (cf. $Q' \propto 1/\sqrt{\lambda_0 + \lambda_T}$), and the decreasing rate at which it does so (for increasing $\lambda_0$), corresponding to the flow being completely blocked, leads to convexity and a corresponding increase in expected power ($\partial^2 P_{GC05}/\partial \lambda_0^2 > 0$). For a laterally unconfined turbine (GC13), the flow rate through the farm initially increases with increasing background friction (cf. $u_T/u_0 = 2\lambda_0/(\lambda_T + 2\lambda_0)$, but must do so at a decreasing rate (for increasing $\lambda_0$), because the flow through the farm cannot be stopped, leading to concavity and a corresponding decrease in expected power ($\partial^2 P_{GC05}/\partial \lambda_0^2 < 0$). Examining figure 4b once more, as the number of turbines relative to the background friction ($\lambda_T/\lambda_0$) is increased, a transition occurs from concavity ($\partial^2 P_{GC13}/\partial \lambda_0^2 < 0$) associated with a reduction in expected power to convexity ($\partial^2 P_{GC13}/\partial \lambda_0^2 > 0$) associated with an increase in expected power. We will refer to $\lambda_T < 4\lambda_0$ as background friction dominated and $\lambda_T > 4\lambda_0$ as turbine friction dominated.

### 2.2.2. Optimal turbine drag

An analytical expression for the optimal turbine drag $\lambda_{T\text{stoch}}^*$ can be found by maximizing (2.16) with respect to $\lambda_T$ such that

$$\lambda_{T\text{stoch}}^* = 2\mu_{\lambda 0}\left[1 + \frac{1}{2}\hat{\sigma}_{\lambda_0}^2\right] + \mathcal{O}(E[\Delta\lambda_0{}^3]) \quad \text{with } \hat{\sigma}_{\lambda_0} = \frac{\sigma_{\lambda_0}}{\mu_{\lambda 0}}. \tag{2.20}$$

Compared to the downward shift in optimal turbine drag in response to background friction uncertainty for a fully spanned channel (2.6) (GC05), which was only diminished in magnitude by the effect of inertia (V10), the optimal turbine drag in a laterally unconfined channel is shifted upwards (see figure 4b). This can be explained by alluding to (2.7), noting that $P_{\lambda_T\lambda_T} < 0$ for optimum power. From figure 4a it is evident that at the deterministic optimum $\lambda_T/\lambda_0 = 2$ about which we perturb, increasing the number of turbines ($\lambda_T$) acts to reduce the concavity of the power with respect to background friction

**Table 1.** Leading-order effects of uncertainty in background friction ($\hat{\sigma}_{\lambda_0} = \sigma_{\lambda_0}/\mu_{\lambda_0}$) on the relative change in expected power, the relative standard deviation of power and the relative change in optimal turbine friction in drag-dominated (CG05) and inertia-dominated (V10) fully spanned channels and in a laterally unconfined farm (GC13).

| | change in expected power $(E[P] - P_{det})/P_{det}$ | standard deviation in power $\sigma_P/P_{det}$ | optimal turbine friction $\Delta\lambda_T^*/\lambda_{T,det}^*$ |
|---|---|---|---|
| drag-dominated fully spanned channel (GC05) | $\frac{15}{8}\left(1 + \frac{\lambda_T}{\mu_{\lambda_0}}\right)^{-2}\hat{\sigma}_{\lambda_0}^2$ | $\frac{3}{2}\left(1 + \frac{\lambda_T}{\mu_{\lambda_0}}\right)^{-1}\hat{\sigma}_{\lambda_0}$ | $-\frac{5}{6}\hat{\sigma}_{\lambda_0}^2$ |
| fully spanned channel (V10) | negative for $\mu_{\lambda_0} < 0.495$, positive for $\mu_{\lambda_0} > 0.495$ (see appendix A) | smaller than GC05 | negative, but smaller in magnitude than GC05 |
| laterally unconfined farm (GC13) | $-\frac{(\lambda_T/\mu_{\lambda_0})(4-\lambda_T/\mu_{\lambda_0})}{(\lambda_T/\mu_{\lambda_0}+2)^2}\hat{\sigma}_{\lambda_0}^2$ | $2\frac{\lambda_T}{\mu_{\lambda_0}}(2 + \frac{\lambda_T}{\lambda_{C_0}})^{-1}\hat{\sigma}_{\lambda_0}$ | $+\frac{1}{2}\hat{\sigma}_{\lambda_0}^2$ |

($P_{\lambda_0\lambda_0\lambda_T} > 0$ at $\lambda_T/\lambda_0 = 2$) and thus $\Delta\lambda_T^* > 0$ from (2.7). It is optimal to move more into the turbine friction-dominated regime.

### 2.2.3. Uncertainty in power

The variance in power to leading order is given by

$$\sigma_P^2 = \frac{4\lambda_T^4\mu_{\lambda_0}^2}{(\lambda_T + 2\mu_{\lambda_0})^6}\sigma_{\lambda_0}^2 + \mathcal{O}(E[\Delta\lambda_0^3]) = (P_{GC13}(\lambda_0 = \mu_{\lambda_0}))^2\frac{4(\lambda_T/\mu_{\lambda_0})^2}{(2 + (\lambda_T/\mu_{\lambda_0}))^2}\hat{\sigma}_{\lambda_0}^2, \tag{2.21}$$

which is illustrated in figure 5. It is evident from this figure and (2.18) that increasing turbine friction as a share of mean background friction ($\lambda_T/\mu_{\lambda_0}$) increases the variability in power, tending towards a constant multiple of $\hat{\sigma}_{\lambda_0}$ as $\lambda_T/\mu_{\lambda_0} \to \infty$.

## 2.3. Comparison of models

Table 1 summarizes the effects of background friction uncertainty on expected power, optimal turbine drag, and power uncertainty for a fully spanned channel in the quasi-steady limit (GC05) and with inertia (V10), as well as for a laterally unconfined turbine farm (GC13). It is evident from this table that bottom friction uncertainty acts to increase the expected power in a fully spanned channel, but generally has an opposite effect in laterally unconfined farms. The optimal number of turbines with bottom friction uncertainty is lower in a fully spanned channel and higher in laterally unconfined farms. Bypass flow fundamentally changes how the system behaves under uncertainty. In fully spanned channels, inertia acts to reduce the effect of uncertainty in background friction (V10 versus GC05).

## 3. Calibration of bottom friction uncertainty

In order to quantify its effect on power, we must estimate the magnitude of background friction uncertainty in the form of the relative standard deviation $\hat{\sigma}_{\lambda_0} = \sigma_{\lambda_0}/\mu_{\lambda_0}$. In each of the foregoing models, $\lambda_0$ is simply a linear function of the respective bottom roughness coefficients, ignoring the effect of exit separation. Consequently, we can set the relative standard deviations to be equal: $\sigma_{\lambda_0}/\mu_{\lambda_0} = \sigma_{C_0}/\mu_{C_0}$, where $\mu_{C_0}$ is the mean and $\sigma_{C_0}$ the standard deviation of the bottom roughness coefficient. With *a priori* knowledge of both the tidal elevation and flow rate of a channel, the bed roughness coefficient could be relatively accurately determined from the phase difference between the two. With the exception of measurement errors, the uncertainty associated with background friction would be small, provided the flow conditions and thus the background friction experienced are not substantially altered by the introduction of turbines. In the absence of knowledge of both the tidal elevation and the flow rate, $C_0$ is essentially unknown, as an observed elevation may be the result of an enormous number of combinations of bed roughness coefficients and flow rates. Equally, various values of tidal elevation and bed roughness may be combined to give an observed flow rate. Data

**Table 2.** Parametric uncertainty resulting from variation in bed roughness length $z_0$ for different bed conditions. The number of values reported for different bed conditions and the resulting mean and variation factor are taken from [18]. The standard deviation of the natural logarithm of $z_0$ is given by the logarithm of the variation factor v.f. ($\sigma_{\ln z_0} = \ln(\text{v.f.})$). From these values and using the properties of the log-normal distribution, we compute the (relative) standard deviation of bed roughness length $\hat{\sigma}_{z_0}$ ($\sigma_{z_0}/\mu_{z_0}$).

| bed material and type | no. of values reported | $\mu_{z_0}$ (mm) | variation factor, v.f. | $\sigma_{z_0}$ (mm) | $\sigma_{z_0}/\mu_{z_0}$ |
|---|---|---|---|---|---|
| mud | 1 | 0.2 | — | — | — |
| mud/sand | 3 | 0.7 | 4.1 | 1.8 | 2.5 |
| silt/sand | 1 | 0.05 | — | — | — |
| unrippled sand | 7 | 0.4 | 2.0 | 0.3 | 0.8 |
| rippled sand | 6 | 6.0 | 1.3 | 1.6 | 0.3 |
| sand/shell | 2 | 0.3 | 4.5 | 0.9 | 2.9 |
| sand/gravel | 7 | 0.3 | 6.7 | 1.8 | 6.0 |
| mud/sand/gravel | 2 | 0.3 | 3.0 | 0.5 | 1.5 |
| gravel | 4 | 3 | 1.6 | 1.5 | 0.5 |

on tidal elevation are often available. However, volumetric flux is usually far more difficult to determine. Point measurements of current velocities are sometimes available from acoustic doppler current profiler (ADCP) deployments and help confine the possible values of bed roughness coefficient to a region.

We distinguish two sources of background friction uncertainty. First, the roughness length parameter $Z_0$ captures the magnitude of the friction coefficient at a site, which is dependent on the bed material and type and may be unknown, as well as vary across a given site. We will refer to the probability that the roughness length $Z_0$ is smaller than or equal to a value $z_0$, namely $\Pr(Z_0 \leq z_0)$, as arising from parametric uncertainty. Second, for a known value of the roughness length parameter $z_0$, many different models predict different friction coefficients $C_0$. We will refer to the probability that the predicted friction coefficient $C_0$ is smaller than or equal to a value $c_0$ for a known value of the roughness length parameter $z_0$, namely $\Pr(C_0 \leq c_0 \mid Z_0 = z_0)$, as arising from model uncertainty, which is conditional on parametric uncertainty. The unconditional probability $\Pr(C_0 \leq c_0)$ then describes the likelihood that a particular value of $c_0$ correctly captures the bed shear stress due to the flow and it results from the two underlying sources of uncertainty, which we will estimate separately below. In the following, capital variables refer to random variables, while lower-case variables to specific values that the random variables may take.

## 3.1. Parametric uncertainty: $\Pr(\hat{Z}_0 \leq \hat{z}_0)$

The bed friction coefficient at a site is usually expressed as a function of relative roughness $\hat{Z}_0 = Z_0/h$, where $Z_0$ is the roughness length associated with a particular bed material and type and $h$ is the water depth. Several phenomena contribute to the roughness length. These are the skin friction, due to the surface roughness of the sediment grains of the bed; the form drag, caused by the pressure field due to the presence of larger bed features; and the sediment-transport contribution, produced by momentum transfer of the flow to mobilized sediment particles [8]. An additional component relating to vegetation contributes in cases where there is plant growth at the bed. These components are commonly assumed to combine linearly to give the total roughness length. In absence of velocity profile measurements at a site, the bed roughness length may be estimated from knowledge of the bed conditions. The uncertainty in $Z_0$, then, stems from the difficulty in defining a single value for the roughness length due to spatial heterogeneity of the seafloor, variation in bed-grain sizes, change of bed forms with time (e.g. sand dunes travelling with the flow), as well as dependence of $Z_0$ on the hydrodynamic regime (i.e. whether the flow is hydrodynamically rough, smooth, or transitional).

In order to obtain an estimate for uncertainty in relative roughness $\hat{Z}_0$, we specifically consider the skin friction component of the bed roughness length. Table 2 lists values for the roughness lengths, obtained by fitting logarithmic velocity profiles for a range of different bed conditions, taken from [18]. For seven of the nine bed conditions listed, Soulsby [18] reports the geometric mean and variation factor obtained from a number of values reported in the literature. From these values, shown

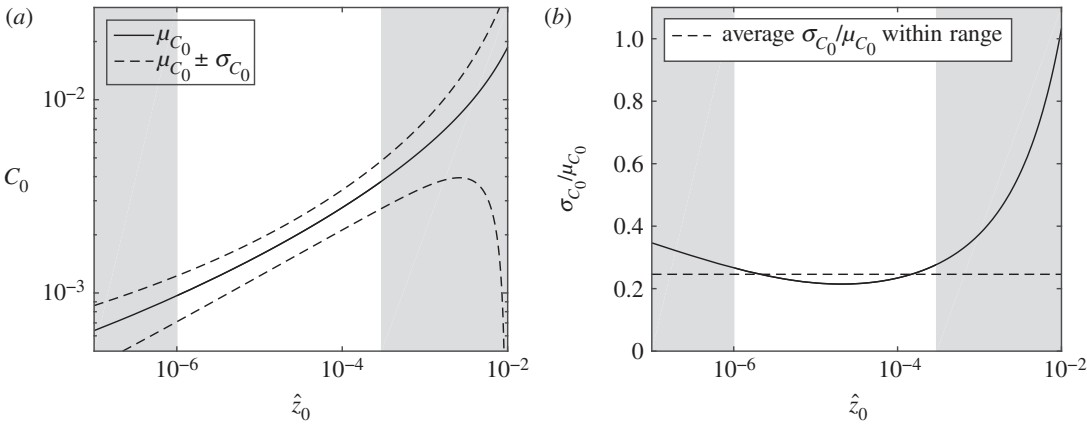

**Figure 7.** (a) Model uncertainty based on eight different methods to determine the drag coefficient $C_0$ as a function of relative roughness $\hat{z}_0$, showing the average $\mu_{C_0}(\hat{z}_0)$ (continuous black) and one standard deviation either side $\mu_{C_0}(\hat{z}_0) \pm \sigma_{C_0}(\hat{z}_0)$ (dashed black) as a function of $\hat{z}_0$. (b) Relative standard deviation in $C_0$ as a function of $\hat{z}_0$ (continuous) and the average (dashed line) for the range of consideration. Shaded areas denote values of relative roughness beyond the limits of $\hat{z}_0$ relevant for tidal energy, i.e. $\hat{z}_0 = 1 \times 10^{-6}$ as the lower limit and $\hat{z}_0 = 3 \times 10^{-4}$ as the upper limit.

in table 2, we compute the standard deviation $\sigma_{z_0}$ and the relative standard deviation $\sigma_{z_0}/\mu_{z_0}$. It is evident that the uncertainty here is very considerable. Furthermore, the relative standard deviation depends strongly on bed type and ranges from 0.5 for gravel to 6.0 for a sand/gravel mixture. Because finer grains fill gaps between coarser grains, beds made up of a mixture of grain sizes have relatively low roughness lengths [18], while also exhibiting a higher standard deviation because the degree of filling will probably vary greatly according to the proportions of the different grain sizes present. This may be seen from the values in table 2, where the relative standard deviation is typically larger for bed type mixtures than for beds made up of a single type.

When considering how to apply results such as those shown in table 2 to a site, several scenarios in terms of available information and associated uncertainty are possible. Of these we consider two limiting scenarios. The first scenario is where accurate knowledge of the bed conditions exists, such that the relevant value of relative standard deviation $\sigma_{z_0}/\mu_{z_0}$ in the final column of table 2 may be used. This we consider a lower limit on uncertainty, identical to that of conditional model uncertainty in §3.2. In the second scenario, the bed conditions may be entirely unknown or might vary across a site. Assuming this latter limit, which forms a more realistic estimate, we proceed in a somewhat *ad hoc* fashion and assign equal probabilities to each of the bed conditions in table 2 except for *mud* (only a single value reported), *silt/sand* (only a single value reported) and *rippled sand* (includes components of form drag and hence omitted) to estimate the relative standard deviation as follows:

$$\hat{\sigma}_{z_0} = \frac{\sqrt{\sum_i w_i \sigma_{z_0,i}^2}}{\sum_i w_i \mu_{z_0,i}}, \tag{3.1}$$

where the subscript $i$ corresponds to a row in table 2, and weights are assigned according to the number of values reported (from [18]) so that $\sum_i w_i = 1$. From this, we obtain the large value of $\hat{\sigma}_{z_0} = 1.6$ and use this as our base case. Ignoring uncertainty in the water depth, we set $\hat{\sigma}_{\hat{z}_0} = \hat{\sigma}_{z_0} = 1.6$

## 3.2. Model uncertainty: $\Pr(C_0 \leq c_0 | \hat{Z}_0)$

To estimate uncertainty resulting from the application of different friction coefficient models for a known value of the roughness length $\hat{z}_0$, we consider the eight different $C_0$-models summarized in fig. 13 of [19] (reproduced in figure 7a). The eight empirical models are derived from fitting experimental data to either a power-law relationship of the form $C_0 = \alpha \hat{z}_0^{\beta}$ or a logarithmic law of the form $C_0 = [\kappa/(B + \ln \hat{z}_0)]^2$, where $\kappa$ is von Kármán's constant. Table 3 in appendix B lists these two commonly used, empirical formulae for estimating $C_0$ (left-hand column) and the values of the parameters $\alpha$, $\beta$, $B$ and $\kappa$ fitted from experimental and numerical data by different authors. We take an agnostic approach and assign equal weights to each of the eight models to determine the mean friction coefficient $\mu_{C_0}$ and the standard

rsos.royalsocietypublishing.org    R. Soc. open sci. **6**: 180941

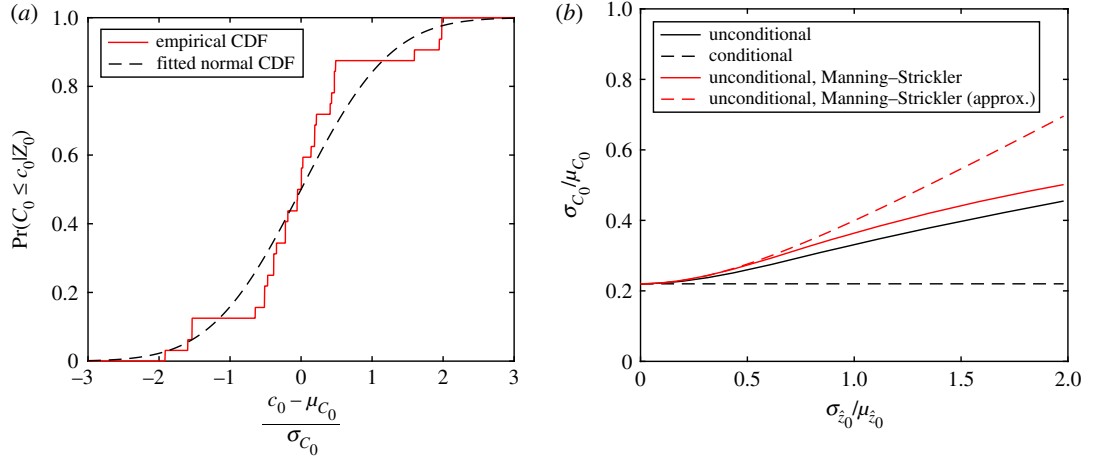

**Figure 8.** Empirical cumulative distribution function for conditional model uncertainty (a) and variation of the relative standard deviation of the unconditional uncertainty in $C_0$ with the relative standard deviation of parametric uncertainty at a mean relative roughness of $\mu_{\hat{z}_0} = 1.51 \times 10^{-4}$ (b).

deviation $\sigma_{C_0}$ across a range of values for $\hat{z}$ known with certainty. It is evident from figure 7a that model uncertainty is considerable.

In particular, we are interested in the behaviour of the relative standard deviation across $\hat{z}_0$ values that are appropriate for tidal stream energy assessments. A lower bound on the $\hat{z}_0$ range in tidal channels is found by dividing the smallest roughness length, that for silt/sand ($z_0 = 5 \times 10^{-5}$ m), by a value of water depth typical for deep channels of approximately 50 m, giving a value of $\hat{z}_{lower} \approx 1 \times 10^{-6}$. An upper bound is found by dividing the largest value for $z_0$ (that for rippled sand, $z_0 = 6 \times 10^{-3}$ m) by a typical lower value for water depth of approximately 20 m, thus giving a value of $\hat{z}_{upper} \approx 3 \times 10^{-4}$. By considering the relative standard deviation throughout this (unshaded) range in figure 7b, it can be shown that this property has a weak dependence on the value of $\hat{z}_0$. The relative standard deviation $\hat{\sigma}_{C_0} \equiv \sigma_{C_0}/\mu_{C_0}$ varies between a minimum value of 0.21 and a maximum of 0.28, with an average value of 0.25 (indicated as a dashed line in figure 7b). At the midpoint of the range considered, $\hat{z}_0 = 1.51 \times 10^{-4}$, the value for the relative standard deviation is 0.22. A normal, non-skewed, distribution is appropriate for model uncertainty, which is evident from figure 8a, which shows the empirical cumulative distribution function. This distribution is estimated by creating a sample population from selecting four (arbitrary, yet equally spaced) relative roughness values of $\hat{z} = [10^{-6}, 10^{-5}, 10^{-4}, 10^{-3}]$, scaled by their local means and standard deviations, as shown in figure 8a.

## 3.3. Unconditional uncertainty: $\Pr(C_0 \le c_0)$

To address the scenario in which the bed conditions are not known or vary across a site, we combine parametric uncertainty with conditional model uncertainty from the previous sections, in order to obtain the unconditional uncertainty. Motivated by [18], we use a log-normal probability distribution to capture parametric uncertainty and set its mean $\mu_{\hat{z}_0}$ equal to the average of the lower and upper bounds for $\hat{z}_0$ relevant to tidal energy determined earlier, namely $\mu_{\hat{z}_0} = 1.51 \times 10^{-4}$, for different values of $\hat{\sigma}_{z_0}$. We numerically convolve the log-normal parametric uncertainty distribution with the eight equally weighted $C_0$-models from the previous section and calculate statistical moments. Figure 8b shows the unconditional relative standard deviation $\hat{\sigma}_{C_0}$ as a function of relative standard deviation of relative roughness length $\hat{\sigma}_{\hat{z}_0}$ (continuous black line), the latter as a measure of parametric uncertainty. The conditional relative standard deviation $\hat{\sigma}_{C_0}$ is shown as a horizontal dashed black line and corresponds to the value of 0.22 obtained in the previous section.

In the case of a log-normal distribution for $\hat{z}_0$ and for the Manning–Strickler formula $C_0 = \alpha \hat{z}_0^{\beta}$ (see appendix B), convolution may be achieved analytically. Assuming that $\alpha$ and $\hat{z}_0$ are independent random variables, the variance in $C_0$ is given generally by

$$\mathrm{Var}[C_0] = \mathrm{Var}[\alpha]\mathrm{Var}[\hat{z}_0^{\beta}] + \mathrm{Var}[\alpha]E[\hat{z}_0^{\beta}]^2 + E[\alpha]^2\mathrm{Var}[\hat{z}_0^{\beta}], \tag{3.2}$$

rsos.royalsocietypublishing.org   R. Soc. open sci. 6: 180941

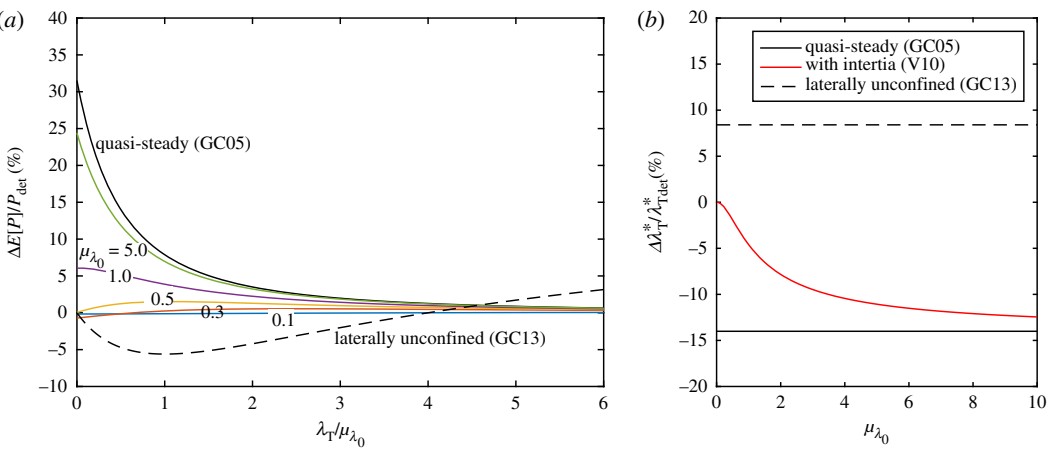

**Figure 9.** Quantitative estimates of relative change in expected power due to background friction ($\hat{\sigma}_{\lambda_0} = 0.41$) as a function of turbine drag-scaled with mean background friction at different values of $\mu_{\lambda_0}$ (a) and relative change in optimal turbine friction in the presence of uncertainty in the background friction parameter (b) for a fully spanned tidal channel (GC05 and V10).

which, by dividing by $E[C_0]^2$, may be expressed in terms of relative variances,

$$\hat{\sigma}^2_{C_0,\text{uncond}} = \hat{\sigma}^2_{C_0,\text{cond}}[\hat{\sigma}^2_{\hat{z}_0^\beta} + 1] + \hat{\sigma}^2_{\hat{z}_0^\beta}, \tag{3.3}$$

where $\hat{\sigma}^2_{C_0,\text{uncond}} \equiv \text{Var}[C_0]/E[C_0]^2$ denotes the unconditional variance of $C_0$, $\hat{\sigma}^2_{C_0,\text{cond}} \equiv \text{Var}[\alpha]/E[\alpha]^2$ the variance in $C_0$ conditional on $\hat{z}_0$, and $\hat{\sigma}^2_{\hat{z}_0^\beta} \equiv \text{Var}[\hat{z}_0^\beta]/E[\hat{z}_0^\beta]^2$ is the relative variance of a power-law function of the uncertain bottom friction parameter. For a log-normally distributed $\hat{z}_0$, we have exactly $\hat{\sigma}^2_{\hat{z}_0^\beta} = (1 + \hat{\sigma}^2_{\hat{z}_0})^{\beta^2} - 1$, which is shown in figure 8b as the continuous red line. This line shows good agreement with the unconditional variance from numerically exact convolution (continuous black line); apparent disagreements are due to models of alternative form also being included in the curve (cf. appendix B). For small values of uncertainty in $\hat{z}_0$ (parametric uncertainty) and $\alpha$ (model uncertainty), (3.3) may be approximated to give

$$\hat{\sigma}^2_{C_0,\text{uncond}} = \hat{\sigma}^2_{C_0,\text{cond}} + \beta^2 \hat{\sigma}^2_{\hat{z}_0}, \tag{3.4}$$

where only leading-order terms are considered in both relative variances and their products are ignored. The dashed red line in figure 8b shows that (3.4) accurately represents (3.3), except for large values of $\hat{\sigma}_{\hat{z}_0}$.

At our base case value for parametric uncertainty of $\hat{\sigma}_{\hat{z}_0} = 1.6$ (derived from (3.1) and table 2), we obtain an estimate for the unconditional relative standard deviation of $\hat{\sigma}_{C_0} = 0.41$ from figure 8b. We use this value $\hat{\sigma}_{\lambda_0} = \hat{\sigma}_{C_0} = 0.41$ in the next section to estimate the quantitative impact of bed roughness uncertainty. We emphasize our estimates are indicative, not definite.

# 4. Quantitative estimates of the effect of uncertainty

## 4.1. Expected power

Figure 9a shows the change in expected power as a percentage of deterministic power for our base case value of relative background friction uncertainty $\hat{\sigma}_{\lambda_0} = 0.41$. For a fully spanned channel dominated by channel drag (GC05), such as a shallow, long channel with $\mu_{\lambda_0} \gg 1$, the increase in expected power can be as large as 30%. In fact, in the limit of very few turbines ($\lambda_T$), we have $(E[P] - P_{\text{det}})/P_{\text{det}} = (15/8)\hat{\sigma}^2_{\lambda_0} \approx 32\%$ (cf. table 1). However, in a deeper channel representative of the Pentland Firth ($\mu_{\lambda_0} = 1.0$, see Vennell _et al._ [15]) the increase in expected power would only be of the order of a few per cent (6%) and would tend to reduce as more turbines are added. For laterally unconfined channels, the effects are negative and generally small (less than 5–10%).

## 4.2. Optimal turbine drag

Figure 9b shows the change in optimal turbine drag as a percentage of the deterministic optimum for $\hat{\sigma}_{\lambda_0} = 0.41$. The change is between −14% for a drag-dominated fully spanned channel (GC05) and

rsos.royalsocietypublishing.org    R. Soc. open sci. 6: 180941

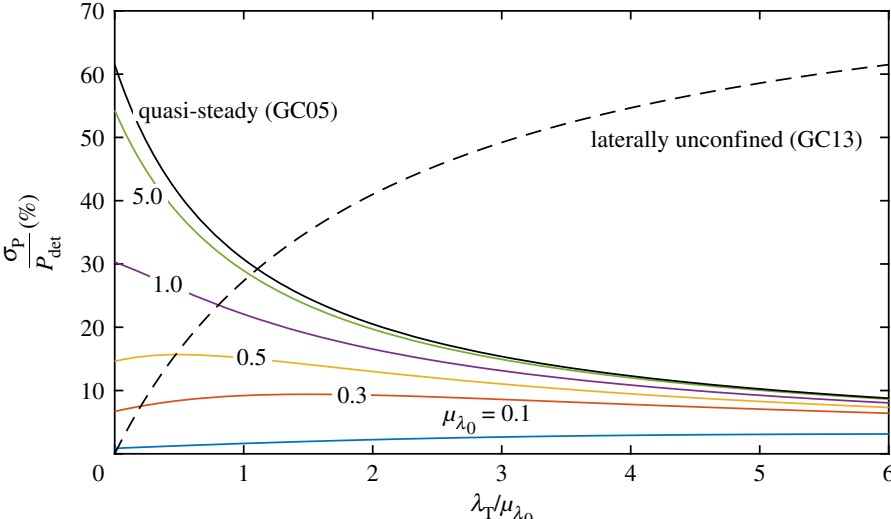

**Figure 10.** Quantitative estimates of relative standard deviation in power due to background friction ($\hat{\sigma}_{\lambda_0} = 0.41$) as a function of turbine drag scaled mean with mean background friction and at different values of $\mu_{\lambda_0}$ for the laterally unconfined farm (V10).

+8% for a laterally unconfined channel (GC13). Inertia acts to reduce the decrease in optimal turbine drag for fully spanned channels; in a channel representative of the Pentland Firth ($\mu_{\lambda_0} = 1.0$), we estimate a reduction in the optimal turbine drag of 5%.

## 4.3. Uncertainty in power

Figure 10 shows the standard deviation of power as a percentage of the deterministic power for $\hat{\sigma}_{\lambda_0} = 0.41$. For drag-dominated fully spanned channel (GC05) with few turbines, the relative standard deviation reaches 62%. Again, inertia reduces this. In a channel representative of the Pentland Firth ($\mu_{\lambda_0} = 1.0$), we estimate a relative standard deviation of 30%.

## 5. Conclusion

Estimates of the tidal power that can be extracted at a given site are subject to significant uncertainty, with different estimates sometimes more than an order of magnitude apart. Of the many sources of uncertainty, uncertainty in bed friction can be considerable—both due to unknown and spatially varying bed conditions and variation in the predictions of different bed friction models. To illustrate this, we estimate the parametric uncertainty resulting from lack of knowledge of bed conditions at a particular site, to be associated with a relative standard deviation of $\hat{\sigma}_{\hat{z}_0} = 1.6$, if we assign equal probabilities to a range of commonly occurring bed types. Even with precise knowledge of the bed conditions, we estimate a relative standard deviation of 0.22 associated with the range of predictions for $C_0$ from different models outlined in table 3. We combine these uncertainties to give an unconditional uncertainty in bed roughness of $\hat{\sigma}_{C_0} = 0.41$ (one relative standard deviation) related to a typical site for tidal turbine deployment. This estimate constitutes a lower limit for uncertainty in the bed roughness coefficient at a particular site for the data presented in table 2, because it is assumed that the bed conditions (and their variability) are known. In reality, this knowledge is unlikely, and so the uncertainty in $C_0$ is likely to be greater. Furthermore, spatio-temporal variability in bed conditions, which is not discussed here, will act to increase the value for $\hat{\sigma}_{C_0}$. For a given site, the uncertainty may be constrained by performing appropriate seabed surveys and a better estimate for bed roughness coefficient may be found, though uncertainty will remain.

In order to make a quantitative assessment of the effect of background friction uncertainty on estimates of tidal power potential, we have incorporated such uncertainty in three idealized models of tidal energy extraction, of which each captures a different element of the key physics. In Garrett & Cummins [10] (GC05), an analytic solution is derived for the power potential of a channel in the drag-dominated limit and fully spanned by turbines. Vennell [11] (V10) relaxes this limit by retaining inertia in the governing equation for a fully spanned channel and derives an analytic solution for power to an approximate form of the governing equation. Finally, Garrett & Cummins [12] (GC13) allow for bypass flows by

considering a laterally unconfined turbine farm. In particular, we have used perturbation methods to derive leading-order estimates for the effect of uncertainty in the value of bed roughness coefficient on three key quantities for each of the models: expected power, standard deviation in power and optimal turbine drag. In the presence of background friction uncertainty and nonlinearity in the model, evaluating power for the expected value of background friction does not give the same answer as evaluating the expectation of power for the distribution of values of background friction (cf. Jensen's inequality). It is the difference between the two that we consider when we compare expected power with deterministic power (evaluated at the mean value of background friction). A similar issue is encountered in wind energy assessment, where the median is used as a measure of power under uncertainty because it is invariant under monotonic nonlinear transformations. Evidently, power is now a random variable and we also consider the standard deviation of its distribution as a relevant measure to understand the confidence we have in tidal resource estimates. Finally, the turbine drag chosen to optimize expected power is different from that chosen to optimize deterministic power.

Our conclusions are as follows. First, for fully spanned channels (GC05 and V10), we have identified two regimes. In the drag-dominated regime, the expected power is larger than the deterministic power, whereas in the inertia-dominated regime the opposite is true. Inertia has the effect of bounding the flow rate at low values of bed roughness, such that the increase in expected power is smaller and even reversed at sufficiently low total channel drag (background+turbine). For channels in which the flow may be diverted around the turbines (GC13), the expected power always decreases, except for extremely large turbine drag (i.e. very many turbines installed). Quantitatively, we estimate expected power can increase by as much as 32% for drag-dominated, quasi-steady channels, which are typically shallow and long, while reducing expected power by only 6% in laterally unconfined flow. In a channel representative of the Pentland Firth ($\mu_{\lambda_0} = 1.0$), the increase in expected power may only be of the order of a few per cent.

Second, uncertainty in power is only enhanced compared to background uncertainty for very drag-dominated fully spanned channels (and for laterally unconfined channels with very large turbine drag). Inertia has the effect of reducing power uncertainty, because power becomes less sensitive to bottom drag in the presence of inertia, and variation in bed roughness produces a relatively smaller variation in power. Laterally unconfined channels behave in the opposite way: for low values of turbine drag, less of the flow is diverted around the farm and the flow rate tends towards a constant, independent of bed roughness. For a channel representative of the Pentland Firth, the uncertainty in extractable power may be as large as 30% (one relative standard deviation) for small-scale turbine deployments. This value increases to over 50% for a small, high flow-rate channel ($\mu_{\lambda_0} = 4.5$), indicating that while the shift in expected power resulting from considering uncertainty may be negligible, variation in this power can be considerable. For example, the 95% confidence interval for the power from the Pentland Firth due to uncertainty in bed roughness will be at $\pm 2\hat{\sigma}_P = \pm 60\%$ of the mean power value determined. For a mean power of 5 GW (the mean of the range 0.62–9 GW given in the Introduction) then, the range of likely values for power estimates from the Pentland Firth is 2–8 GW, spanning a significant portion of the range of reported values. However, it must be emphasized that the reported estimates are taken from different models, with different physical assumptions, containing sources of uncertainty other than bed friction (the focus of the present paper) and which may contribute to a greater extent to the range of mean power estimates reported above.

Third, the turbine drag that maximizes expected power in the presence of background uncertainty is greater compared to its deterministic value for laterally unconfined channels (GC13) and smaller for fully spanned channels (GC05), with uncertainty reducing the size of this effect (V10). Generally, however, this effect is small (between −8 and +14%).

There are a number of limitations to these findings. First, the models considered herein are idealized and do not take into account the complex bathymetry of actual tidal sites and associated flow curvature, the complexity of the tidal forcing components or deformation of the free surface. This limits the extent to which the findings from the models may be applied to real sites that exhibit such features. Furthermore, the models used are depth-averaged and so provide suitable power estimates only for regional-scale energy extraction by large turbine deployments [1]. Future work, using a numerical model applied to a real site such as the Pentland Firth (e.g. that of Adcock *et al.* [5]), would take these into account and could thus be used to validate the predictions made in this paper. Second, we consider here only the effect on extractable power, which does not take into account mixing in the wake of the turbines, instead of available power. Future work, would apply the methodology of the present paper to linear momentum actuator disc theory [20] to include the effect of wake mixing. Third, we have only considered uncertainty in background friction and not turbine drag itself. Future work would

consider uncertainty regarding the correct value for enhanced bed roughness to use in a depth-averaged model to capture accurately the thrust exerted by rows of turbines [21,22].

Data accessibility. This work does not have primary data. Code used to numerically determine the optimal turbine drag for V10 is available from the Dryad Digital Repository: https://doi.org/10.5061/dryad.cp7v85c [23].
Authors' contributions. M.J.K. and T.S.v.d.B. conceived the mathematical models, with computations performed by M.J.K.; the mathematical models were extended to include the V10 tidal energy model by S.D.; results were interpreted by M.J.K., T.S.v.d.B. and S.D.; A.G.L.B. informed the bed roughness uncertainty calibration; M.J.K. and T.S.v.d.B. wrote the paper. All authors gave final approval for publication.
Competing interests. We declare we have no competing interests.
Funding. M.J.K. was supported by Engineering and Physical Sciences Research Council (EPSRC) grant no. R44708. T.S.v.d.B. was supported by a Royal Academy of Engineering Research Fellowship.
Acknowledgements. All contributors have been included as authors. The authors would like to express their thanks to the three anonymous reviewers for their insightful comments which greatly helped to improve the original manuscript.

# Appendix A. Statistical moments for V10

## A.1. Expected power

The expected power for V10 is determined in the same way as that for GC05. Power in V10 (2.11) is expanded in terms of a Taylor series in $\Delta\lambda_0$ about the deterministic case $\lambda_0 = \mu_{\lambda_0}$, truncated to second order. We use the shorthand $\lambda_{eq} \equiv 8(\lambda_0 + \lambda_T)/(3\pi)$ and $\tilde{\mu} \equiv \sqrt{4\mu_{\lambda_{eq}}^2 + 1}$ to reduce clutter. The Taylor expansion is done in terms of $\lambda_{eq}$, i.e. $\lambda_{eq} = \mu_{\lambda_{eq}} + \Delta\lambda_{eq}$ where $\mu_{\lambda_{eq}} = 8(\mu_{\lambda_0} + \lambda_T)/(3\pi)$ and $\Delta\lambda_{eq} = 8\Delta\lambda_0/(3\pi)$. Noting that the variance of the equivalent channel drag is $\sigma_{\lambda_{eq}}^2 = (8\sigma_{\lambda_0}/(3\pi))^2$ due to the linear transformation, the expected power is given by

$$
\frac{1}{P_0}E[P_{V10}] = \frac{4}{3\pi}\lambda_T\frac{(\tilde{\mu}-1)^{3/2}}{(\sqrt{2}\mu_{\lambda_{eq}})^3}
$$
$$
+ \frac{4}{3\pi}\lambda_T\frac{3(10\mu_{\lambda_{eq}}^4(\tilde{\mu}-4) + \mu_{\lambda_{eq}}^2(19\tilde{\mu}-27) + 4(\tilde{\mu}-1))}{2\sqrt{2}\mu_{\lambda_{eq}}^5(4\mu_{\lambda_{eq}}^2+1)^{3/2}\sqrt{\tilde{\mu}-1}}\sigma_{\lambda_{eq}}^2 + \mathcal{O}(E[\Delta\lambda_{eq}^3]). \tag{A 1}
$$

The first term is simply the power calculated from the V10 model (2.11) at a drag of $\lambda_{eq} = \mu_{\lambda_{eq}}$. The second term indicates the leading-order response of the model to uncertainty. The change in expected power as a fraction of deterministic power $(E[P_{V10}] - P_{det})/P_{det}$ changes sign from negative to positive at a value of $\mu_{\lambda_{eq}} = 0.420$, i.e. $\mu_{\lambda_0} + \lambda_T = 0.495$.

## A.2. Variance

In a similar manner, the variance for V10, i.e. $\sigma_{P_{V10}}^2 = E[P_{V10}^2] - E[P_{V10}]^2$, is given by

$$
\sigma_{P_{V10}}^2 = \lambda_T^2\frac{8(1 - \tilde{\mu} + (7 - 5\tilde{\mu})\mu_{\lambda_{eq}}^2 + (13 - 5\tilde{\mu})\mu_{\lambda_{eq}}^4 + 4\mu_{\lambda_{eq}}^6)}{\pi^2\tilde{\mu}^3\mu_{\lambda_{eq}}^8} + \mathcal{O}(E[\Delta\lambda_{eq}^3]). \tag{A 2}
$$

## A.3. Optimum turbine drag

The turbine drag which maximizes the expected power (A 1) was found numerically by using a Newton–Raphson algorithm to find the value of $\lambda_T$ (in the limit of a small standard deviation in $\lambda_0$, to be consistent with the expansions in the other sections) which satisfies $\partial(A 1)/\partial\lambda_T = 0$ and gave a negative second derivative.

# Appendix B. Bed roughness coefficient models

Table 3 lists several commonly encountered formulae for bed roughness coefficients, derived from empirical and numerical experiments, as a function of roughness length $z_0$. The formulae are shown in figure 11 to illustrate the spread in values of $C_0$ for a given value of roughness length. The mean bed roughness coefficient $\mu_{C_0}$ and one standard deviation $\pm\sigma_{C_0}$ either side are superimposed onto the models.

rsos.royalsocietypublishing.org    R. Soc. open sci. **6**: 180941

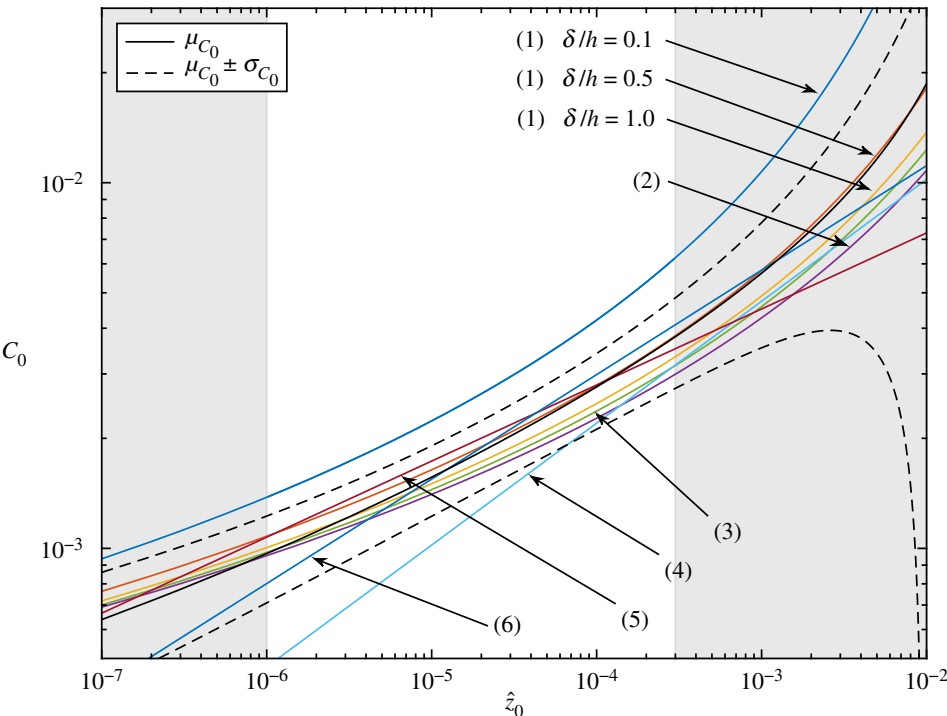

**Figure 11.** Model uncertainty based on eight different methods (numbered according to table 3) to determine the drag coefficient $C_0$ as a function of relative roughness $\hat{z}_0$, showing the average $\mu_{C_0}(\hat{z}_0)$ (continuous black) and one standard deviation either side $\mu_{C_0}(\hat{z}_0) \pm \sigma_{C_0}(\hat{z}_0)$ (dashed black) as a function of $\hat{z}_0$. Shaded regions denote values of relative roughness beyond the limits for $\hat{z}_0$ relevant for tidal energy, i.e. $\hat{z}_0 = 1 \times 10^{-6}$ as lower and $\hat{z}_0 = 3 \times 10^{-4}$ as upper limit.

**Table 3.** Eight different formulae for calculating bed roughness coefficient $C_0$, used to estimate model uncertainty for given relative roughness $\hat{z}_0 = z_0/h$, taken from [19]. Here $\kappa$ is von Kármán's constant and its values and those of the parameters $\alpha$, $\beta$ and $B$ have been obtained from experimental and numerical data by different authors.

| formula | label in figure 11 and name | parameters |
|---|---|---|
| $C_0 = \left[ \frac{\kappa}{B + \ln(\hat{z}_0)} \right]^2$ | (1) Deep water [19] | $\kappa = 0.40$ |
| | boundary-layer thickness $\delta$ | $B = (\delta/2h) - \log(\delta/2h)$ |
| | (2) Colebrook–White [24] | $\kappa = 0.405$ |
| | $z_0 = (k_s/30) + (\nu/9u_*)$ | $B = 0.71$ |
| | (3) full-depth logarithmic [19] | $\kappa = 0.40$ |
| | velocity profile | $B = 1$ |
| $C_0 = \alpha \hat{z}_0^{\beta}$ | (4) Manning–Strickler [19] | $\alpha = 0.0474$ |
| | | $\beta = \frac{1}{3}$ |
| | (5) Dawson–Johns [25] | $\alpha = 0.0190$ |
| | | $\beta = 0.208$ |
| | (6) Soulsby [18] | $\alpha = 0.0415$ |
| | | $\beta = \frac{2}{7}$ |

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
