## [Reviewer comments · Royal Society Open Science]

Review History

RSOS-180941.R0 (Original submission)

Review form: Reviewer 1 (Clym Stock-Williams)

Is the manuscript scientifically sound in its present form?

Yes

Are the interpretations and conclusions justified by the results?

Yes

Is the language acceptable?

Yes

Is it clear how to access all supporting data?

No

Do you have any ethical concerns with this paper?

No

Have you any concerns about statistical analyses in this paper?

No

Recommendation?

Accept as is

Comments to the Author(s)

This is an excellent paper. The analysis is clear and in-depth.

There are some very minor comments which the authors may wish to consider:

- Figure 7 is difficult to read in colour, and impossible in Black & White.
- The reason why the median (P50) is used in wind energy to provide energy output estimates is because it is invariant under monotonic non-linear transformations, while the mean, as you point out, is not.

Review form: Reviewer 2 (Peter Stansby)

Is the manuscript scientifically sound in its present form?

Yes

Are the interpretations and conclusions justified by the results?

Yes

Is the language acceptable?

Yes

Is it clear how to access all supporting data?

Not Applicable

Do you have any ethical concerns with this paper?

No

Have you any concerns about statistical analyses in this paper?

No

Recommendation?

Major revision is needed (please make suggestions in comments)

Comments to the Author(s)

This paper analyses the effect of uncertainty of bed friction factor on power generation from tidal turbines based on three idealised models: Garrett and Cummins (2005,2013) and Vennell(2010). These models assume that flow is straight, parallel with fully developed boundary layer across the depth. There appears to have been some comparisons with depth-averaged models which require the input of a bed friction coefficient. The depth-averaged assumption is inherent in the three idealised models. There is discussion on the magnitude of bed friction factor and its relation to bed roughness. There is no discussion on whether the boundary layer is fully developed. An estimate of boundary layer thickness δ for oscillatory flows is given by $\delta / k_N = 0.09 (a/k_N)^{0.82}$ where a is particle oscillation amplitude and k_N is Nikuradse roughness (about twice

roughness height), see Fredsoe and Deigaard (1994). For velocity amplitude of 3 m/s and tidal period of 12 hours δ is 61 m for $kN=0.0001$ m (silt in table 2 which is lower limit). Flow velocity magnitude is not actually mentioned but I understand velocities need to be at least 3 m/s for tidal turbines to be effective. δ is thus likely to be greater than depth (50 m for deep channels) but it is worth checking. More important flow curvature affects bed friction generally increasing its magnitude. For wakes (of both headlands and islands) this may be by an order of magnitude, e.g. Stansby et al (2016). I understand the Pentland Firth comprises headlands and islands. Curvature can also cause significant secondary flows which affect dispersion and hence mixing. Curvature may be due to bathymetry or wakes resulting from turbine interaction. Oceanographers use 3-D models with hydrostatic pressure as a matter of practice. In the local wake region bed friction coefficient is also likely to be increased due to enhanced turbulence mixing but this may be regarded as part of the uncertainty in turbine induced drag. (In isolation thrust coefficient does not give a good wake representation in a depth-averaged model.) The point is that the analysis presented is valid for boundary layers developed through the depth and almost parallel flows (implying widely spaced turbines). There is the question whether the idealised models have been compared with physical experiments or detailed 3-D models as this would help the justification of this analysis. I do not see this mentioned although the models are widely quoted; there is certainly data available to do this. Assuming this can be addressed there is a case for publication. The analysis seems quite sound and novel albeit quite dense and some figures can take a little time to follow. However the conditions for which this analysis might apply in practice need to be addressed.

Finally, what is Jensen's inequality?

Fredsoe, J. and Deigaard, R. 1994 Mechanics of coastal sediment transport, Advanced Series on Ocean Engineering - Vol.3, World Scientific, Singapore.

Stansby, P. Chini, N. and Lloyd, P. 2016 Oscillatory flows around a headland by 3D modelling with hydrostatic pressure and implicit bed shear stress comparing with experiment and depth-averaged modelling, Coastal Engineering 116, 1-14

Review form: Reviewer 3

Is the manuscript scientifically sound in its present form?

Yes

Are the interpretations and conclusions justified by the results?

Yes

Is the language acceptable?

Yes

Is it clear how to access all supporting data?

Not Applicable

Do you have any ethical concerns with this paper?

No

Have you any concerns about statistical analyses in this paper?

I do not feel qualified to assess the statistics

Recommendation?

Accept with minor revision (please list in comments)

Comments to the Author(s)

The reviewer would like to congratulate the authors for a very good paper. The following comments, particularly comments 8, 9 and 11, could improve the paper in my opinion:

- 1- Sentence starting at page 3 line 32 is not clear.
- 2- Page 3 Line 49, It will be useful to highlight more clearly why these scenarios have been selected in the introduction. One may think that more realistic scenarios, particularly in terms of scenarios 1 and 2, could be more meaningful.
- 3- Sentence in page 5 line 20 is not clear.
- 4- Statement in Page 6 line 36 seems to be the case for only small turbine drag parameter. Could you clarify please?
- 5- sentence in Page 6 line 46 needs more explanation.
- 6- Page 8 line 35 what is Q' .
- 7- Page 9 line 37. Is derivation of an expression for the expected power in V_{10} , namely $E[PV_{10}]$, published anywhere like a thesis? it might be helpful for the readers to have reference or include them as an appendix.
- 8- Page 15 line 53. In reviewer's opinion, the values reported in Table 2 are not the only correct values of roughness for different bed materials. Even if the bed roughness is known and constant across the domain, the standard deviation could vary significantly comparing to the values calculated here based on the Table 2. This is further highlighted by limited availability for some cases such as Mud and Silt/sand in the table. Therefore, considering two different scenarios and particularly one scenario where the values are considered as known is not accurate and should change to highlight wide range of uncertainty in roughness if the writers keep two scenarios.
- 9- Page 18 line 34. It is useful to discuss various assumptions made and to what level these assumptions could impact the results to avoid misunderstanding by various stakeholders.
- 10- Page 20 line 10. range of predictions of what?
- 11- Page 20 line 54 (last line of page 20). There are other sources of uncertainty and it might not be accurate to associate big part of the variation in reported values to bed roughness uncertainty. It is suggested that the author's look into the models used in these references and make sure that the values used for roughness correspond to these extremes before associating major part of uncertainty on these numbers.

Decision letter (RSOS-180941.R0)

27-Sep-2018

Dear Ms Kreitmair,

The editors assigned to your paper ("The effect of uncertain bottom friction on estimates of tidal current power") have now received comments from reviewers. We would like you to revise your paper in accordance with the referee and Associate Editor suggestions which can be found below (not including confidential reports to the Editor). Please note this decision does not guarantee eventual acceptance.

Please submit a copy of your revised paper before 20-Oct-2018. Please note that the revision deadline will expire at 00.00am on this date. If we do not hear from you within this time then it will be assumed that the paper has been withdrawn. In exceptional circumstances, extensions may be possible if agreed with the Editorial Office in advance. We do not allow multiple rounds of revision so we urge you to make every effort to fully address all of the comments at this stage. If deemed necessary by the Editors, your manuscript will be sent back to one or more of the

original reviewers for assessment. If the original reviewers are not available, we may invite new reviewers.

- Data accessibility

If you wish to submit your supporting data or code to Dryad (<http://datadryad.org/>), or modify your current submission to dryad, please use the following link:
<http://datadryad.org/submit?journalID=RSOS&manu=RSOS-180941>

- Competing interests

- Authors' contributions

- Acknowledgements

- Funding statement

Please note that Royal Society Open Science charge article processing charges for all new submissions that are accepted for publication. Charges will also apply to papers transferred to Royal Society Open Science from other Royal Society Publishing journals, as well as papers submitted as part of our collaboration with the Royal Society of Chemistry (<http://rsos.royalsocietypublishing.org/chemistry>). If your manuscript is newly submitted and subsequently accepted for publication, you will be asked to pay the article processing charge, unless you request a waiver and this is approved by Royal Society Publishing. You can find out more about the charges at <http://rsos.royalsocietypublishing.org/page/charges>. Should you have any queries, please contact openscience@royalsociety.org.

Kind regards,

Royal Society Open Science Editorial Office
Royal Society Open Science
openscience@royalsociety.org

on behalf of Prof. R. Kerry Rowe (Subject Editor)
openscience@royalsociety.org

Associate Editor's comments:

The referees see much of merit in your work, but they each have some concerns (ranging from minor to major issues) that need to be addressed prior to further consideration. Most notably, concerns were raised that key code for the models is lacking. As the journal requires that authors submit their data, code or other digital research materials (http://rsos.royalsocietypublishing.org/author-information#Open_data), you must provide this in any revision.

Comments to Author:

Reviewers' Comments to Author:

Reviewer: 1

Comments to the Author(s)

This is an excellent paper. The analysis is clear and in-depth.

There are some very minor comments which the authors may wish to consider:

- Figure 7 is difficult to read in colour, and impossible in Black & White.
- The reason why the median (P50) is used in wind energy to provide energy output estimates is because it is invariant under monotonic non-linear transformations, while the mean, as you point out, is not.

Reviewer: 2

Comments to the Author(s)

This paper analyses the effect of uncertainty of bed friction factor on power generation from tidal turbines based on three idealised models: Garrett and Cummins (2005,2013) and Vennell(2010). These models assume that flow is straight, parallel with fully developed boundary layer across the depth. There appears to have been some comparisons with depth-averaged models which require the input of a bed friction coefficient. The depth-averaged assumption is inherent in the three idealised models. There is discussion on the magnitude of bed friction factor and its relation to bed roughness. There is no discussion on whether the boundary layer is fully developed. An estimate of boundary layer thickness δ for oscillatory flows is given by $\delta / k_N = 0.09 (a/k_N)^{0.82}$ where a is particle oscillation amplitude and k_N is Nikuradse roughness (about twice roughness height), see Fredsoe and Deigaard (1994). For velocity amplitude of 3 m/s and tidal period of 12 hours δ is 61 m for $k_N=0.0001$ m (silt in table 2 which is lower limit). Flow velocity magnitude is not actually mentioned but I understand velocities need to be at least 3 m/s for tidal turbines to be effective. δ is thus likely to be greater than depth (50 m for deep channels) but it is worth checking. More important flow curvature affects bed friction generally increasing its magnitude. For wakes (of both headlands and islands) this may be by an order of magnitude, e.g. Stansby et al (2016). I understand the Pentland Firth comprises headlands and islands. Curvature can also cause significant secondary flows which affect dispersion and hence mixing. Curvature may be due to bathymetry or wakes resulting from turbine interaction. Oceanographers use 3-D models with hydrostatic pressure as a matter of practice. In the local wake region bed friction coefficient is also likely to be increased due to enhanced turbulence mixing but this may be regarded as part of the uncertainty in turbine induced drag. (In isolation thrust coefficient does not give a good wake representation in a depth-averaged model.) The point is that the analysis presented is valid for boundary layers developed through the depth and almost parallel flows (implying widely spaced turbines). There is the question whether the idealised models have been compared with physical experiments or detailed 3-D models as this would help the justification of this analysis. I do not see this mentioned although the models are widely quoted; there is certainly data available to do this. Assuming this can be addressed there is a case for publication. The analysis seems quite sound and novel albeit quite dense and some figures can take a little time to follow. However the conditions for which this analysis might apply in practice need to be addressed.

Finally, what is Jensen's inequality?

Fredsoe, J. and Deigaard, R. 1994 Mechanics of coastal sediment transport, Advanced Series on Ocean Engineering - Vol.3, World Scientific, Singapore.

Stansby, P. Chini, N. and Lloyd, P. 2016 Oscillatory flows around a headland by 3D modelling with hydrostatic pressure and implicit bed shear stress comparing with experiment and depth-averaged modelling, Coastal Engineering 116, 1-14

Reviewer: 3

Comments to the Author(s)

The reviewer would like to congratulate the authors for a very good paper. The following comments, particularly comments 8, 9 and 11, could improve the paper in my opinion:

- 1- Sentence starting at page 3 line 32 is not clear.

2- Page 3 Line 49, It will be useful to highlight more clearly why these scenarios have been selected in the introduction. One may think that more realistic scenarios, particularly in terms of scenarios 1 and 2, could be more meaningful.

3- Sentence in page 5 line 20 is not clear.

4- Statement in Page 6 line 36 seems to be the case for only small turbine drag parameter. Could you clarify please?

5- sentence in Page 6 line 46 needs more explanation.

6- Page 8 line 35 what is Q' .

7- Page 9 line 37. Is derivation of an expression for the expected power in V_{10} , namely $E[PV_{10}]$, published anywhere like a thesis? it might be helpful for the readers to have reference or include them as an appendix.

8- Page 15 line 53. In reviewer's opinion, the values reported in Table 2 are not the only correct values of roughness for different bed materials. Even if the bed roughness is known and constant across the domain, the standard deviation could vary significantly comparing to the values calculated here based on the Table 2. This is further highlighted by limited availability for some cases such as Mud and Silt/sand in the table. Therefore, considering two different scenarios and particularly one scenario where the values are considered as known is not accurate and should change to highlight wide range of uncertainty in roughness if the writers keep two scenarios.

9- Page 18 line 34. It is useful to discuss various assumptions made and to what level these assumptions could impact the results to avoid misunderstanding by various stakeholders.

10- Page 20 line 10. range of predictions of what?

11- Page 20 line 54 (last line of page 20). There are other sources of uncertainty and it might not be accurate to associate big part of the variation in reported values to bed roughness uncertainty. It is suggested that the author's look into the models used in these references and make sure that the values used for roughness correspond to these extremes before associating major part of uncertainty on these numbers.

Author's Response to Decision Letter for (RSOS-180941.R0)

See Appendix A.

RSOS-180941.R1 (Revision)

Review form: Reviewer 1 (Clym Stock-Williams)

Is the manuscript scientifically sound in its present form?

Yes

Are the interpretations and conclusions justified by the results?

Yes

Is the language acceptable?

Yes

Is it clear how to access all supporting data?

Yes

Do you have any ethical concerns with this paper?

No

Have you any concerns about statistical analyses in this paper?

No

Recommendation?

Accept as is

Comments to the Author(s)

Thank you for your considered responses to all comments. I believe this paper is now a valuable contribution to the literature,

Review form: Reviewer 2 (Peter Stansby)

Is the manuscript scientifically sound in its present form?

Yes

Are the interpretations and conclusions justified by the results?

Yes

Is the language acceptable?

Yes

Is it clear how to access all supporting data?

Not Applicable

Do you have any ethical concerns with this paper?

No

Have you any concerns about statistical analyses in this paper?

No

Recommendation?

Accept with minor revision (please list in comments)

Comments to the Author(s)

The comments have been answered well, with one exception below, and the response to other referees has also enhanced the paper. The contribution is well defined and original. However one important condition for the analysis is that oscillatory tidal flow corresponds to fully developed steady channel flow with an associated friction coefficient. A reader may get the idea from this paper that this is universally valid and of course oscillatory boundary layers are different from steady boundary layers. Some justification based on the boundary layer thickness, e.g. Fredsoe and Deigaard (1994), being greater than the water depth should be included in my opinion. In the response the authors give an analysis of the effect of a power law velocity profile on the depth-averaged quantities. Incidentally the analysis looks rather similar to that in Liggett's classic text book. The power law relates to fully developed steady channel flow and that this occurs requires some justification.

Decision letter (RSOS-180941.R1)

01-Nov-2018

Dear Ms Kreitmair:

On behalf of the Editors, I am pleased to inform you that your Manuscript RSOS-180941.R1 entitled "The effect of uncertain bottom friction on estimates of tidal current power" has been accepted for publication in Royal Society Open Science subject to minor revision in accordance with the referee suggestions. Please find the referees' comments at the end of this email.

The reviewers and Subject Editor have recommended publication, but also suggest some minor revisions to your manuscript. Therefore, I invite you to respond to the comments and revise your manuscript.

- Ethics statement

- Data accessibility

If you wish to submit your supporting data or code to Dryad (<http://datadryad.org/>), or modify your current submission to dryad, please use the following link:
<http://datadryad.org/submit?journalID=RSOS&manu=RSOS-180941.R1>

- Competing interests

- Authors' contributions

- Acknowledgements

- Funding statement

Because the schedule for publication is very tight, it is a condition of publication that you submit the revised version of your manuscript before 10-Nov-2018. Please note that the revision deadline will expire at 00.00am on this date. If you do not think you will be able to meet this date please let me know immediately.

Supplementary files will be published alongside the paper on the journal website and posted on

the online figshare repository (<https://figshare.com>). The heading and legend provided for each supplementary file during the submission process will be used to create the figshare page, so please ensure these are accurate and informative so that your files can be found in searches. Files on figshare will be made available approximately one week before the accompanying article so that the supplementary material can be attributed a unique DOI.

Please note that Royal Society Open Science charge article processing charges for all new submissions that are accepted for publication. Charges will also apply to papers transferred to Royal Society Open Science from other Royal Society Publishing journals, as well as papers submitted as part of our collaboration with the Royal Society of Chemistry (<http://rsos.royalsocietypublishing.org/chemistry>). If your manuscript is newly submitted and subsequently accepted for publication, you will be asked to pay the article processing charge, unless you request a waiver and this is approved by Royal Society Publishing. You can find out more about the charges at <http://rsos.royalsocietypublishing.org/page/charges>. Should you have any queries, please contact openscience@royalsociety.org.

on behalf of Prof. R. Kerry Rowe (Subject Editor)
openscience@royalsociety.org

Associate Editor Comments to Author:

While the referees consider your paper to be almost ready for publication, there are a couple of remaining issues requiring attention. Please note reviewer 2's comments, in particular in relation to the similarity of this work to earlier publications: please ensure it is made clear how your work is an meaningful contribution.

Reviewer comments to Author:

Reviewer: 1

Comments to the Author(s)

Thank you for your considered responses to all comments. I believe this paper is now a valuable contribution to the literature,

Reviewer: 2

Comments to the Author(s)

The comments have been answered well, with one exception below, and the response to other referees has also enhanced the paper. The contribution is well defined and original. However one important condition for the analysis is that oscillatory tidal flow corresponds to fully developed steady channel flow with an associated friction coefficient. A reader may get the idea from this paper that this is universally valid and of course oscillatory boundary layers are different from steady boundary layers. Some justification based on the boundary layer thickness, e.g. Fredsoe and Deigaard (1994), being greater than the water depth should be included in my opinion.

In the response the authors give an analysis of the effect of a power law velocity profile on the depth-averaged quantities. Incidentally the analysis looks rather similar to that in Liggett's classic text book. The power law relates to fully developed steady channel flow and that this occurs requires some justification.

Author's Response to Decision Letter for (RSOS-180941.R1)

See Appendix B.

Decision letter (RSOS-180941.R2)

09-Nov-2018

Dear Ms Kreitmair,

I am pleased to inform you that your manuscript entitled "The effect of uncertain bottom friction on estimates of tidal current power" is now accepted for publication in Royal Society Open Science.

on behalf of Prof R. Kerry Rowe (Subject Editor)
openscience@royalsociety.org

Follow Royal Society Publishing on Twitter: [@RSocPublishing](https://twitter.com/RSocPublishing)
Follow Royal Society Publishing on Facebook:
<https://www.facebook.com/RoyalSocietyPublishing.FanPage/>
Read Royal Society Publishing's blog: <https://blogs.royalsociety.org/publishing/>

Appendix A

RSOS-180941

Title: The effect of uncertain bottom friction on estimates of tidal current power

Authors: M.J. Kreitmair, S. Draper, A.G.L. Borthwick, and T.S. van den Bremer

Journal: Royal Society Open Science

Dear Editor,

Thank you very much for your kind email concerning our manuscript titled “The effect of uncertain bottom friction on estimates of tidal current power” (Manuscript ID RSOS-180941) which was submitted to *Royal Society Open Science* for possible publication. We have modified the paper taking full consideration of all the comments and suggestions from the Reviewers. Corresponding revisions are indicated by red font in the new version of our manuscript. We also provide a detailed list of the revisions, along with itemized responses to each comment and suggestion, which we believe have led to a significant improvement in the quality of the manuscript. We confirm that all the format, style and referencing style of the manuscript should fit the requirements of the Journal.

We would like to thank you again for your detailed suggestions. Sincere thanks are also due to the anonymous reviewers for their very helpful comments.

With best regards.

Sincerely yours,

Monika Kreitmair

+++++

Authors' Response

Associate Editor's Comments:

The referees see much of merit in your work, but they each have some concerns (ranging from minor to major issues) that need to be addressed prior to further consideration. Most notably, concerns were raised that key code for the models is lacking. As the journal requires that authors submit their data, code or other digital research materials (http://rsos.royalsocietypublishing.org/author-information#Open_data), you must provide this in any revision.

Response: Thank you for your kind comments regarding the paper, which are much appreciated. The only part of our results that cannot be explicitly computed from the equations given in the paper, and required solving using numerical methods, is the optimum value of turbine drag parameter for the model of Vennell (2010). We have appended the methodology for finding this as Appendix A to the document, along with the explicit equations for expected power and variance to make the derivations clearer. We have also added our Matlab scripts (written by the first author) to Dryad, as requested. The DOI for the code is 10.5061/dryad.cp7v85c. Finally, we have addressed each concern raised by the referees, as indicated below.

Reviewer #1 Comments

This is an excellent paper. The analysis is clear and in-depth.

Response: Thank you for your comments regarding the paper, which are much appreciated.

There are some very minor comments which the authors may wish to consider:

- Figure 7 is difficult to read in colour, and impossible in Black & White.

Response: We agree with the reviewer that the figure was difficult to read and decided to remove the coloured curves in Figure 7, moving them to a new figure 11 in Appendix B alongside the information used to produce the figure. We hope this has made the figures easier to follow.

- The reason why the median (P50) is used in wind energy to provide energy output estimates is because it is invariant under monotonic non-linear transformations, while the mean, as you point out, is not.

Response: This is a very useful comment. We have now included a brief comment on this in the second paragraph of the Conclusion section which reads

“A similar issue is encountered in wind energy assessment, where the median is used as a measure of power under uncertainty because it is invariant under monotonic nonlinear transformations.”

Reviewer #2 Comments

This paper analyses the effect of uncertainty of bed friction factor on power generation from tidal turbines based on three idealised models: Garrett and Cummins (2005,2013) and Vennell(2010). These models assume that flow is straight, parallel with fully developed boundary layer across the depth. There appears to have been some comparisons with depth-averaged models which require the input of a bed friction coefficient. The depth-averaged assumption is inherent in the three idealised models. There is discussion on the magnitude of bed friction factor and its relation to bed roughness. There is no discussion on whether the boundary layer is fully developed. An estimate of boundary layer thickness δ for oscillatory flows is given by $\delta/k_N = 0.09 (a/k_N)^{0.82}$ where a is particle oscillation amplitude and k_N is Nikuradse roughness (about twice roughness height), see Fredsoe and Deigaard (1994). For velocity amplitude of 3 m/s and tidal period of 12 hours δ is 61 m for $k_N=0.0001$ m (silt in table 2 which is lower limit). Flow velocity magnitude is not actually mentioned but I understand velocities need to be at least 3 m/s for tidal turbines to be effective. δ is thus likely to be greater than depth (50 m for deep channels) but it is worth checking.

Response: We are grateful to the reviewer for this insightful comment. Using the empirical formula provided by Fredsøe and Deigaard, and noting that $k_N \approx 30z_0$ where z_0 is the roughness height (taken to be 0.05 mm), we obtain a very similar result that δ is 99 m for typical conditions. For a typical deep channel, the boundary layer is larger than the water depth.

We would like to point out that the governing equations used in GC05, V10, and GC13 are based on the one-dimensional depth-averaged shallow water equations, which are essentially the same as the St Venant equations used in river flow modelling but with the effect of the side walls of the domain neglected – a reasonable assumption in wide channels. During the depth-averaging process, the non-uniform velocity profile through the boundary layer can be accounted for, and appears as a correction to the uniform (rather than boundary layer) flow momentum exchange. Amongst others, Falconer (1985) provides a detailed derivation which highlights this aspect and may be outlined as follows. The x -momentum equation after depth integration (neglecting shear stress terms, though this is not necessary for the derivation) can be written as

$$\int_{-h_s}^{\zeta} \left[\frac{\partial \bar{u}}{\partial t} + \frac{\partial \bar{u}^2}{\partial x} + g \frac{\partial \zeta}{\partial x} \right] dz = 0, \quad (1)$$

where $\bar{u}(x, z, t)$ is the local horizontal velocity component as a function of the vertical elevation z and time t . Expanding the integrals using Leibnitz' rule, defining the depth-averaged velocity component as

$$U = \frac{1}{h} \int_{-h_s}^{\zeta} \bar{u} dz, \quad (2)$$

where $h = h_s + \zeta$, and substituting for \bar{u} in terms of the depth-averaged velocity in the convective acceleration term as $\bar{u} = U + (\bar{u} - U)$, we obtain

$$\frac{\partial U}{\partial t} + \frac{1}{h} \frac{\partial (hU^2)}{\partial x} + g \frac{\partial \zeta}{\partial x} - \frac{1}{\rho h} \frac{\partial}{\partial x} \int_{-h_s}^{\zeta} [-\rho(\bar{u} - U)^2 - 2\rho U(\bar{u} - U)] dz = 0. \quad (3)$$

Assuming a 1/7th power law,

$$\bar{u} = \frac{8}{7} U \frac{(h_s + z)^{1/7}}{(h_s + \zeta)^{1/7}}, \quad (4)$$

such that

$$\bar{u} - U = \left(\frac{8}{7} \frac{(h_s + z)^{1/7}}{(h_s + \zeta)^{1/7}} - 1 \right) U, \quad (5)$$

giving

$$\int_{-h_s}^{\zeta} [(\bar{u} - U)^2] dz = 0.016hU^2. \quad (6)$$

Rearranging (3) we have

$$\frac{\partial U}{\partial t} + \frac{1}{h} \frac{\partial (1.016hU^2)}{\partial x} + g \frac{\partial \zeta}{\partial x} = 0. \quad (7)$$

This can be written as

$$\frac{\partial U}{\partial t} + \frac{1}{h} \frac{\partial (\beta hU^2)}{\partial x} + g \frac{\partial \zeta}{\partial x} = 0, \quad (8)$$

where β is the momentum exchange correction for non-uniform depth profile. For a turbulent boundary layer flow over rough bed, $\beta = 1.016$. Thus, β may be assumed to be unity. We have amended the paper to include some discussion about this (see below).

Reference:

Falconer, R. A. Residual currents in Port Talbot Harbour: a mathematical model study. *Proc. Instn Civ. Engrs*, Part 2, **79**, pp 33-53, 1985.

More important flow curvature affects bed friction generally increasing its magnitude. For wakes (of both headlands and islands) this may be by an order of magnitude, e.g. Stansby et al (2016). I understand the Pentland Firth comprises headlands and islands. Curvature can also cause significant secondary flows which affect dispersion and hence mixing. Curvature may be due to bathymetry or wakes resulting from turbine interaction. Oceanographers use 3-D models with hydrostatic pressure as a matter of practice. In the local wake region bed friction coefficient is also likely to be increased due to enhanced turbulence mixing but this may be regarded as part of the uncertainty in turbine induced drag. (In isolation thrust coefficient does not give a good wake representation in a depth-averaged model.) The point is that the analysis presented is valid for boundary layers developed through the depth and almost parallel flows (implying widely spaced turbines). There is the question whether the idealised models have been compared with physical experiments or detailed 3-D models as this would help the justification of this analysis. I do not see this mentioned although the models are widely quoted; there is certainly data available to do this. Assuming this can be addressed there is a case for publication. The analysis seems quite sound and novel albeit quite dense and some figures can take a little time to follow. However the conditions for which this analysis might apply in practice need to be addressed.

Fredsoe, J. and Deigaard, R. 1994 *Mechanics of coastal sediment transport*, Advanced Series on Ocean Engineering – Vol.3, World Scientific, Singapore.

Stansby, P. Chini, N. and Lloyd, P. 2016 Oscillatory flows around a headland by 3D modelling with hydrostatic pressure and implicit bed shear stress comparing with experiment and depth-averaged modelling, *Coastal Engineering* 116, 1–14

Response: Thank you for your perceptive comments. We agree that the tidal energy models of Garrett & Cummins (2005, 2013) and Vennell (2010) are highly idealised and assume uniform and parallel flow. However, they nevertheless provide insight into the underlying physics of tidal energy extraction at regional scales. The prediction of GC05 for the maximum power from a channel is in broad agreement with numerical models (see Sutherland *et al.* (2007), Karsten *et al.* (2008), and Adcock *et al.* (2013) who modelled the Pentland Firth), though the other two models do not have similar validation. For large turbine arrays and at large spatial scales, depth-averaged models provide a good estimate of power potential because the bulk flow is adequately captured (see Adcock *et al.*, 2015). We use the models to develop rules of thumb which act as first-order approximations lending insight into how uncertainty propagates through a simple system.

Insight into how changing the physical assumptions in the power models affects uncertainty propagation is developed by considering each of the models in turn. We have added the following text to the Introduction section to make these aims clearer.

“Insight into the effect of the underlying physical assumptions on uncertainty propagation is developed by considering closed-form solutions for power dissipated as predicted by three analytic models of power tidal power assessment. The first model is that of Garrett & Cummins (2005) (henceforth GC05), who derive an analytic solution for quasi-steady flow in a channel spanned completely by tidal turbines. Second, we explore the impact of retaining inertia by examining the solution to the same governing equation by Vennell (2010) (henceforth V10). V10 is able to include inertia in a closed-form solution by making further approximations (see appendix of V10). Third and finally, we examine the effect of flow diversion around the turbines by considering Garrett & Cummins (2013) (henceforth GC13), who consider a circular turbine farm in laterally unconfined flow.”

Furthermore, to clarify the limitations of the tidal models used, and to address the comments on the depth of the boundary layer and flow curvature, we have added the following text to the last paragraph of the Conclusion section

“There are a number of limitations to these findings. First, the models considered herein are idealised and do not take into account the complex bathymetry of actual tidal sites and associated flow curvature, the complexity of the tidal forcing components, or deformation of the free surface. This limits the extent to which the findings from the models may be applied to real sites that exhibit such features. Furthermore, the models used are depth-averaged and so provide suitable power estimates only for regional scale energy extraction by large turbine deployments (Adcock *et al.*, 2015). ”

Finally, we agree that, in practice, a more accurate procedure to assess the impact of uncertainty would be to use two or three-dimensional numerical models to determine how uncertainty propagates from the model input through to the extractable (or indeed available) power. This procedure would take the form of

- Setting up the two or three-dimensional numerical model for a particular site and given bathymetric data.
 - Running the model for a set of bed roughness coefficients and turbine drag values to populate a power surface in these two variables.
 - Define a probability density function for the bed roughness coefficient and transform this through the developed surface to obtain expected power, standard deviation for power, and optimal turbine drag as a function of the turbine drag parameter, mean bed roughness coefficient, and standard deviation.
-
-

The work suggested is outwith the scope of the present paper and represents a different study than the one undertaken herein. However, the authors are currently in the process of writing a companion paper where a two-dimensional numerical model of the Pentland Firth is being used to investigate the ability of the rules of thumb to capture the effect of uncertainty in bed roughness coefficient on power dissipated, utilising the above methodology. To address this, we include a sentence in the last paragraph of the Conclusion section stating “Future work, using a numerical model applied to a real site such as the Pentland Firth (e.g. that of Adcock *et al.* (2013)), would take these [effects] into account and could thus be used to validate the predictions made in this paper.”

References:

- G. Sutherland, M. Foreman, and C. Garrett. Tidal current energy assessment for Johnstone Strait, Vancouver Island. *Proc. Inst. Mech. Eng. A*, 221(2):147–157, 2007.
- R. H. Karsten, J. M. McMillan, M. J. Lickley, and R. D. Haynes. Assessment of tidal current energy in the Minas Passage, Bay of Fundy. *Proc. Inst. Mech. Eng. A*, 222:493–507, 2008.
- T. A. A. Adcock, S. Draper, G. T. Houlsby, A. G. L. Borthwick, and S. Serhadlioglu. The available power from tidal stream turbines in the Pentland Firth, *Proc. R. Soc. A*, 469(2157), 2013.
- T. A. A. Adcock, S. Draper, T. Nishino. Tidal power generation - A review of hydrodynamic modelling, *Proc. I. Mech. E.: A*, 0(0), 2015.

Finally, what is Jensen’s inequality?

Response: Jensen’s inequality is a result from probability theory and, within the context of the discussion in the paper, states that the transformation of a mean through a convex function is less than or equal to the mean applied after the convex transformation. We have added commentary on Jensen’s inequality in section 2.a.i reading

“This convexity results in an asymmetric power dissipation for symmetric perturbations in λ_0 and thus an increase in the expected power (*cf.* Jensen's inequality, which states that a convex transformation of the mean of a random variable is less than the mean of the convex transformation of the variable).”

We hope this makes the reference in the discussion more accessible. See, for example: https://en.wikipedia.org/wiki/Jensen%27s_inequality

Reviewer #3 Comments

The reviewer would like to congratulate the authors for a very good paper.

Response: Thank you for your kind remarks concerning the paper.

The following comments, particularly comments 8, 9 and 11, could improve the paper in my opinion:

1 - Sentence starting at page 3 line 32 is not clear.

Response: We believe this comment refers to page 2, line 32. Based on subsequent references made in this review, all references to pages appear to be shifted in page number by +1. The sentence previously reading

“Adcock *et al.* also found that no single value of C_0 produced results in full agreement with observations, and settled on a value of $C_0 = 0.005$ in a compromise between matching model predictions and field measurements of the tidal phase and the current magnitude for the Pentland Firth.”

has been amended to read

“In addition, Adcock *et al.* found that no single value of C_0 applied throughout the modelled domain produced results which matched the field measurements of both tidal phase and current magnitude for the Pentland Firth, and settled on a value of $C_0 = 0.005$ in a compromise.”

In addition, we have also changed the sentence ending on page 3, line 32 to read

“Using our best estimate for the magnitude of the uncertainty in background roughness coefficient, we provide quantitative estimates of the effects of uncertainty on expected power dissipated and optimal channel design.”

2 - Page 3 Line 49 (we believe this refers to page 2, line 49). It will be useful to highlight more clearly why these scenarios have been selected in the introduction. One may think that more realistic scenarios, particularly in terms of scenarios 1 and 2, could be more meaningful.

Response: This is a good point. The aim of the paper is to develop an intuitive understanding of the propagation of uncertainty from the bed roughness coefficient through to the power estimates. The models introduced in the paper were chosen for analysis because they provide closed-form solutions for power in terms of natural bed roughness and turbine drag parameters. This offers a simple way to capture the key

physics and to understand how the effect of uncertainty is affected by changing model assumptions.

To make this clearer, we have changed the text of the last paragraph of the introduction to include the text:

“Insight into the effect of the underlying physical assumptions on uncertainty propagation is developed by considering closed-form solutions for power dissipated as predicted by three analytic models of power tidal power assessment. The first model is that of Garrett & Cummins (2005) (henceforth GC05), who derive an analytic solution for quasi-steady flow in a channel spanned completely by tidal turbines. Second, we explore the impact of retaining inertia by examining the solution to the same governing equation by Vennell (2010) (henceforth V10). V10 is able to include inertia in a closed-form solution by making further approximations (see appendix of V10). Third and finally, we examine the effect of flow diversion around the turbines by considering Garrett & Cummins (2013) (henceforth GC13), who consider a circular turbine farm in laterally unconfined flow.”

3 - Sentence in page 5 line 20 is not clear.

Response: We believe this comment refers to the sentence beginning with “The average power produced by the turbines...” We have changed the sentence to make clearer the calculation of power dissipated by the turbines by replacing it with

“The power dissipated by the turbines is given by multiplication of the turbine drag term by the mass flow rate, *i.e.* $P = \rho \delta_T |Q| Q^2$, where ρ is the fluid density. The average power extracted by the turbines over a tidal cycle is then $\bar{P} = \rho \delta_T \overline{|Q| Q^2} = \rho (ga)^2 (\gamma \omega)^{-1} \lambda_T \overline{|Q'| Q'^2}$, where the overline notation indicates time-averaging over the tidal period.”

4 - Statement in Page 6 line 36 (page 5, line 36) seems to be the case for only small turbine drag parameter. Could you clarify please?

Response: As may be seen in Figure 2, for both values of mean bed roughness parameter (*i.e.* large and small channels), the dashed ‘uncertain’ lines lie above the continuous ones. While the effect is greatest for small turbine drag parameters, it is still exhibited as the value increases. We have amended the text of the paragraph indicated to read:

“This effect is greatest for $\lambda_T = 2\mu_{\lambda_0}/5$, which maximises the second term in (2.5), but remains positive for all values of λ_T , reducing in strength as λ_T increases (and the effect of background roughness becomes less important).”

5 - Sentence in Page 6 line 46 (page 5, line 46) needs more explanation.

Response: This section was intended to be a pedagogical explanation of Jensen’s inequality which we have made explicit by adding the following sentence at the end of the paragraph.

“This convexity results in an asymmetric power dissipation for symmetric perturbations in λ_0 and thus an increase in the expected power (*cf.* Jensen's inequality, which states that a convex transformation of the mean of a random variable is less than the mean of the convex transformation of the variable).”

6 - Page 8 line 35 (page 7, line 35) what is Q' .

Response: Primed variables refer to non-dimensional variables which were introduced to obtain equation (2.2). To clarify this, reference to (2.2) has now been added in line 35 as well as the statement:

“... where Q' is the non-dimensional flow rate...”

7 - Page 9 line 37. (Page 8, line 37.) Is derivation of an expression for the expected power in V10, namely $E[PV10]$, published anywhere like a thesis? It might be helpful for the readers to have reference or include them as an appendix.

Response: We have added a detailed derivation of the expected power, variance and optimum turbine drag for V10 as an appendix. The derivation will also appear in the first-author’s PhD thesis, which is due to be submitted by the end of 2018.

8 - Page 15 line 53. (Refers to page 14, line 53.) In reviewer's opinion, the values reported in Table 2 are not the only correct values of roughness for different bed materials. Even if the bed roughness is known and constant across the domain, the standard deviation could vary significantly comparing to the values calculated here based on the Table 2. This is further highlighted by limited availability for some cases such as Mud and Silt/sand in the table. Therefore, considering two different scenarios and particularly one scenario where the values are considered as known is not accurate and should

change to highlight wide range of uncertainty in roughness if the writers keep two scenarios.

Response: We agree with the reviewer that the data quoted in Table 2 is by no means exhaustive and, as such, sets a limit on the accuracy of the quantitative estimates. However, we are mainly interested in the variation factors given in Table 2 to obtain a sense of the spread of possible C_0 values at a site. A better way of describing the two scenarios in this section might be “limiting scenarios” for the given data, rather than “possible” scenarios, because the scenario in which the bed conditions are known with certainty is a lower limit. To address this source of possible confusion, we have amended the text of the paragraph immediately preceding equation (3.1) to address this explicitly. The text now reads

“When considering how to apply results such as those shown in Table 2 to a site, several scenarios in terms of available information and associated uncertainty are possible. Of these we consider two limiting scenarios. The first scenario is where accurate knowledge of the bed conditions exists, and the relevant value of relative standard deviation σ_{z_0}/μ_{z_0} in the final column of Table 2 may be used. This we consider a lower limit on uncertainty, identical to that of conditional model uncertainty in subsection (c). In the second scenario, the bed conditions may be entirely unknown or might vary across a site. Assuming this latter limit, which forms a more realistic estimate, we proceed...”

9 - Page 18 line 34. (Refers to page 17, line 34.) It is useful to discuss various assumptions made and to what level these assumptions could impact the results to avoid misunderstanding by various stakeholders.

Response: We believe this is a useful addition and that it might be more appropriately addressed in the Conclusion section. There we have added the following text at the end of the first paragraph

“This estimate constitutes a lower limit for uncertainty in the bed roughness coefficient at a particular site for the data presented in Table 2, because it is assumed that the bed conditions (and their variability) are known. In reality, this knowledge is unlikely, and so the uncertainty in C_0 is likely to be greater. Furthermore, spatio-temporal variability in bed conditions, which is not discussed here, will act to increase the value for $\hat{\sigma}_{C_0}$. For a given site, the uncertainty may be constrained by

performing appropriate seabed surveys and a better estimate for bed roughness coefficient may be found, though uncertainty will remain.”

10 - Page 20 line 10. (Page 19, line 10.) Range of predictions of what?

Response: The sentence has been amended to read “...associated with the range of predictions for C_0 from the different models outlined in Table 3” to make clear what is being referred to.

11 - Page 20 line 54 (last line of page 20) (Refers to page 19). There are other sources of uncertainty and it might not be accurate to associate big part of the variation in reported values to bed roughness uncertainty. It is suggested that the author's look into the models used in these references and make sure that the values used for roughness correspond to these extremes before associating major part of uncertainty on these numbers.

Response: We agree with the reviewer that the large range of values reported for mean power from the Pentland Firth is greatly affected by the choice of model, the physics incorporated, etc., as well as uncertainty in bed friction and other physical/model parameters. For example, the two estimates of 0.62 GW (by ABP MER (2007)) and 9 GW (by MacKay (2008)) use the undisturbed kinetic energy flux to determine the power potential for the Pentland Firth from a given flow velocity but do not include bed roughness in these calculations. The difference in the values reported result from refinement of model mesh, and the use of efficiency coefficients. Given that our paper addresses one significant source of uncertainty, bed friction, but does not deal with all other uncertainties, we have decided to moderate our Conclusion by altering the relevant sentence to read as follows

“For a mean power of 5 GW (the mean of the range 0.62–9 GW given in the Introduction) then, the range of likely values for power estimates from the Pentland Firth is 2–8 GW, spanning a significant portion of the range of reported values. However, it must be emphasised that the reported estimates are taken from different models, with different physical assumptions, containing sources of uncertainty other than bed friction (the focus of the present paper) and which may contribute to a greater extent to the range of mean power estimates reported above.”

References:

Quantification of Exploitable Tidal Energy Resources in UK Waters. Technical report, ABP Marine Environmental Research, 2007.

D. MacKay. Sustainable Energy - without the hot air. UIT Cambridge, 2008.

Appendix B

RSOS-180941

Title: The effect of uncertain bottom friction on estimates of tidal current power

Authors: M.J. Kreitmair, S. Draper, A.G.L. Borthwick, and T.S. van den Bremer

Journal: Royal Society Open Science

Dear Editor,

Thank you very much for your email concerning our manuscript titled “The effect of uncertain bottom friction on estimates of tidal current power” (Manuscript ID RSOS-180941.R1) which was submitted to *Royal Society Open Science* for possible publication. We are delighted to hear the manuscript has been accepted for publication subject to minor revisions, which we have documented below.

We would like to express our sincere thanks to the anonymous reviewers for their kind and helpful comments.

With best regards.

Sincerely yours,

Monika Kreitmair

P.S. While uploading the latex files for this submission, I was informed that the man .tex file would not display in the proof. If there are any problems with the compiling of the document, I will be happy to liaise with the editorial team to help fix the issues.

+++++

Authors' Response

Associate Editor Comments to Author:

While the referees consider your paper to be almost ready for publication, there are a couple of remaining issues requiring attention. Please note reviewer 2's comments, in particular in relation to the similarity of this work to earlier publications: please ensure it is made clear how your work is a meaningful contribution.

Response:

We are very happy to be considered for publication and have addressed the one minor point raised by reviewer 2, as indicated below. We feel that a change to the paper in response to this would only confuse the readership.

We would like to note that the “similarity of this work to earlier publications” noted by the editor in the comments of reviewer 2 refers to an analysis outlined in our previous response to the reviewer, and not to the work in the paper. The analysis is established theory (*i.e.* textbook material). If desired, we would be happy to provide such analysis as an appendix or supplementary material.

Reviewer: 1**Comments to the Author(s)**

Thank you for your considered responses to all comments. I believe this paper is now a valuable contribution to the literature.

Reviewer: 2**Comments to the Author(s)**

The comments have been answered well, with one exception below, and the response to other referees has also enhanced the paper. The contribution is well defined and original.

Response:

Thank you for your very kind comment. We are glad our responses have improved the paper. We address your final comment below.

However one important condition for the analysis is that oscillatory tidal flow corresponds to fully-developed steady channel flow with an associated friction coefficient. A reader may get the idea from this paper that this is universally valid and of course oscillatory boundary layers are different from steady boundary layers. Some justification based on the boundary layer thickness, e.g. Fredsoe and Deigaard (1994), being greater than the water depth should be included in my opinion. In the response the authors give an analysis of the effect of a power law velocity profile on the depth-averaged quantities. Incidentally the analysis looks rather similar to that in Liggett's classic text book. The power law relates to fully developed steady channel flow and that this occurs requires some justification.

Response:

We emphasise that the application of the shallow water equations to tidal flows is standard because the period of tidal oscillations is much greater than the timescale required for boundary layer development. In other words, the problem can be considered steady. Indeed, this assumption is so standard that it is not motivated in the seminal papers by Garrett & Cummins (2005), Vennell (2010), and Garrett & Cummins (2013) upon which our paper is based. It is indeed correct that the analysis included for completeness in our previous response can be found in many textbooks, such as Liggett's.